# D2D: Detector-to-Differentiable Critic for Improved Numeracy in Text-to-Image Generation

## Abstract

Text-to-image (T2I) diffusion models have achieved strong performance in semantic alignment, yet they still struggle with generating the correct number of objects specified in prompts. Existing approaches typically incorporate auxiliary counting networks as external critics to enhance numeracy. However, since these critics must provide gradient guidance during generation, they are restricted to regression-based models that are inherently *differentiable*, thus excluding detector-based models, whose count-via-enumeration nature is *non-differentiable*. To overcome this limitation, we propose **Detector-to-Differentiable** (*D2D*), a novel framework that transforms non-differentiable detection models into differentiable critics, thereby leveraging their superior counting ability to guide numeracy generation. Specifically, we design custom activation functions to convert detector logits into binary indicators, which are then used to optimize the noise prior at inference time with pre-trained T2I models. Our extensive experiments on SDXL-Turbo, SD-Turbo, and Pixart-DMD across four benchmarks of varying complexity (low-density, high-density, and multi-object object scenarios) demonstrate consistent and substantial improvements in object counting accuracy, by up to 13.7%, with minimal degradation in overall image quality and computational overhead.

## 1 Introduction

Diffusion-based text-to-image generative models (Podell et al., 2024; Rombach et al., 2022; Sauer et al., 2025; Chen et al., 2024; 2025b) have achieved promising performance in semantic alignment between synthesized images and text prompts, particularly with recent post-enhancement techniques such as fine-tuning (Clark et al., 2024; Chen et al., 2025a; Yang et al., 2024; Wallace et al., 2024; Black et al., 2024; Xu et al., 2023; Fan et al., 2023) or sampling-based, training-free strategies (Wallace et al., 2023; Eyring et al., 2024; Chung et al., 2024; Chefer et al., 2023). However, even with such advanced alignment techniques, T2I diffusion models continue to struggle at generating exact numbers of objects, even in scenarios with fewer than 10 instances requested. As illustrated in Fig. 1, recent semantic alignment methods, such as ReNO (Eyring et al., 2024), which enhances image alignment with user intent via human preference rewards, fail to synthesize images with the exact number of objects specified in the text input. Motivated by this observation, we tackle the challenge of accurate numeracy generation in this work.

Since vanilla T2I models are not explicitly trained to count, existing methods (Kang et al., 2025; Zafar et al., 2024) introduce auxiliary counting critics to provide additional supervision during generation. These correction signals are propagated to the generative backbone through gradients from the external critics, which restricts current approaches to differentiable, regression-based models such as RCC (Hobley & Prisacariu, 2022) and CLIP-Count (Jiang et al., 2023). However, this design inherently excludes detector-based models, which perform counting via bounding box enumeration. Despite being non-differentiable, such detectors (e.g., OWLv2 (Minderer et al., 2023), YOLOv9 (Wang et al., 2024)) often outperform regression-based counterparts (e.g., RCC (Hobley & Prisacariu, 2022), CLIP-Count (Jiang et al., 2023), CounTR (Chang et al., 2022)) in low-density object scenarios due to their more advanced object localization ability, as illustrated in Fig. 2b. To this end, we propose resolving this bottleneck by converting existing object detectors into differentiable

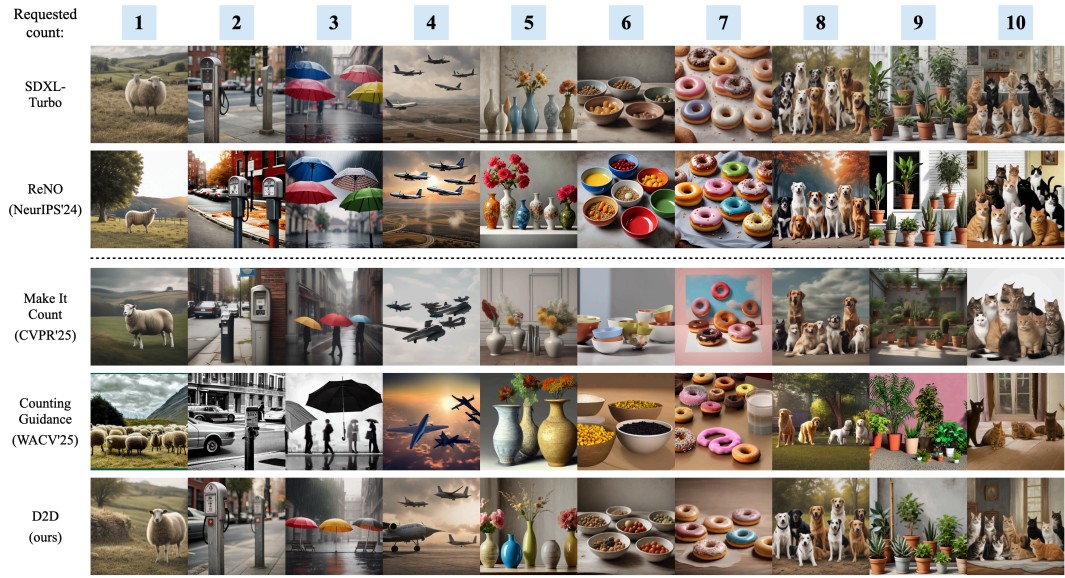

Figure 1: **Qualitative examples illustrating the count-correction ability of our detector-based critic on a variety of objects, counts 1-10.** SDXL-Turbo (Sauer et al., 2025) is a base model with no post-enhancement. ReNO (Eyring et al., 2024) is a generic alignment method, not specifically designed to correct numeracy, that exhibits limited performance in this setting. More recent methods, like Make It Count (Binyamin et al., 2025) and Counting Guidance (Kang et al., 2025), explicitly address count-correction. Our method proposes a new and effective way to leverage detectors for this challenging task. Prompt template: "A realistic photo of a scene with [count] [object class]."

critics, thereby allowing T2I diffusion models to benefit from stronger counting models for improved numeracy.

Our **Detector-to-Differentiable** (*D2D*) framework builds on two key insights that set it apart from existing numeracy-enhancement methods (Kang et al., 2025; Zafar et al., 2024; Binyamin et al., 2025). First, rather than relying on the conventional non-differentiable *"count-via-enumeration"* mechanism, we design a high-curvature activation function that converts bounding box logits outputted from detectors into binary indicators, thereby making them gradient-friendly for count optimization. Second, unlike prior approaches that intervene at intermediate states or denoised predictions along the sampling trajectory, we instead optimize the initial noise using our *"count-via-summation"* gradient. This backbone-agnostic design enables broader generalization of our method across diverse diffusion-based T2I architectures across U-Net (Ronneberger et al., 2015) and DiT (Peebles & Xie, 2023). We further demonstrate the effectiveness of *D2D* via comprehensive experiments using various generative backbones (i.e., SDXL-Turbo (Sauer et al., 2025), SD-Turbo (Sauer et al., 2025), Pixart-DMD Chen et al. (2025b)) and multiple benchmarks (i.e., CoCoCount (Binyamin et al., 2025), D2D-Small, D2D-Multi, D2D-Large), covering diverse numeracy generation scenarios, including single and multiple objects. *D2D* yields the highest numeracy across all multi-step and one-step baselines and benchmarks. In particular, on base model SDXL-Turbo, *D2D* effectively corrects 42% of under-generations (i.e., where the initial generation contains fewer than requested objects) and 40% of over-generations, nearly or more than 2x ReNO (Eyring et al., 2024) and Token Optimization (TokenOpt)'s (Zafar et al., 2024) correction rate. In summary, our contributions are as follows:

- We highlight the importance of accurate numeracy in T2I generation and propose a framework to convert robust object detectors into differentiable critics for count-correction with a newly designed activation function, addressing the bottleneck of having to rely on existing regression-based methods.

- We reposition count-correction problem within the initial noise optimization framework, motivated by the presence of structural priors that exhibit cross-model consistency.

- Our method *D2D* outperforms previous one-step and multi-step count-correction methods by up to **13.7%** points (from 30% with Make It Count to 43.7% with *D2D* on D2D-Small),

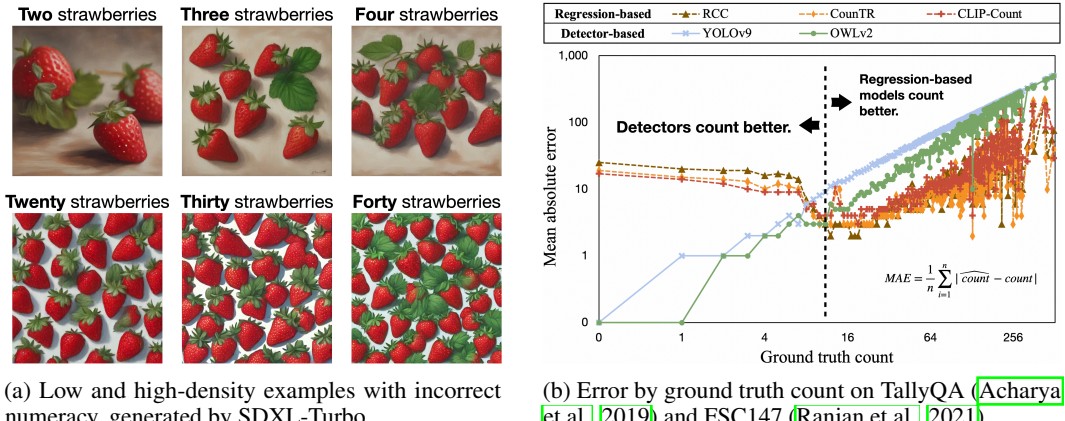

(a) Low and high-density examples with incorrect numeracy, generated by SDXL-Turbo.

(b) Error by ground truth count on TallyQA (Acharya et al., 2019) and FSC147 (Ranjan et al., 2021).

Figure 2: **The low-density setting is where incorrect numeracy is most noticeable and also where detectors count better than regression-based methods.** *But detectors are not differentiable, which precludes them from being used as critics for count correction.*

with minimal degradation in image quality (Fig. 1). On single-object prompts with counts $\leq$ 10, our method introduces less than or comparable computational overhead to baselines.

## 2 RELATED WORK

**Generic alignment enhancement methods.** As noted in the literature (Black et al., 2024; Chen et al., 2025a), the base log-likelihood objective of diffusion models is insufficient on its own to achieve state-of-the-art alignment. To address this, prior works optimize human preference scores via post-enhancement strategies ranging from fine-tuning the U-Net or text encoder (Clark et al., 2024; Xu et al., 2023; Yang et al., 2024; Wallace et al., 2024; Black et al., 2024; Fan et al., 2023; Chen et al., 2025a) to inference-time, training-free strategies that update the intermediate latents (Chung et al., 2024; Chefer et al., 2023). A promising recent line of work (Wallace et al., 2023; Eyring et al., 2024) proposes inference-time alignment via initial noise selection, motivated by the presence of semantic/structural priors in the initial noise (Wang et al., 2025) that influence the semantics/structure of the generated output consistently across diffusion models even with different backbones. But regardless of whether the specific approach is to fine-tune model components or update latents, the problem remains that generic alignment objectives like human-preference scores are insufficient to solve numeracy, as we find there remains a significant gap relative to state-of-the-art count-correction methods like Binyamin et al. (2025). In our work, we specifically address the challenge of improving numeracy with a new formulation for the objective, as well as adopt initial noise optimization as the method of learning, for the ease with which it can be applied across different backbones and the ability to leverage optimized seeds to complement existing methods, as we demonstrate in experiments.

**Numeracy correction methods.** Existing count-correction methods leverage two main mechanisms at inference-time to correct count: (1) apply the gradient of external counting models to correct a tunable portion of the generation process, like Counting Guidance (Kang et al., 2025) and TokenOpt (Zafar et al., 2024), or (2) use attention to control the layout of generated instances, like Make It Count (Binyamin et al., 2025). Counting Guidance uses the RCC counting model (Hobley & Prisacariu, 2022) to optimize the predicted noises, and TokenOpt uses CLIP-Count (Jiang et al., 2023) to optimize the embedding of a count token injected into the prompt as well as a detector to scale down CLIP-Count's overestimates, which increases the computational overhead (about 2-6 times longer than *D2D* on average). Make It Count (Binyamin et al., 2025) is an SDXL-specific (Podell et al., 2024) method that uses self-attention features of the U-Net to extract masks of generated instances and cross-attention to enforce a corrected set of masks. These works are either limited by the need to rely on regression-based counters or manner in which they enforce structure at the cost of image quality, a phenomenon documented in Dinh et al. (2023); Zafar et al. (2024); Patel & Serkh (2025) and noted in our experiments. Instead, *D2D* leverages a more robust *detector-based* critic that enables more effective correction in the low-density setting.

**Regression vs. detector-based counting models.** Regression-based counting methods take an input image and estimate count on a continuous scale. Different variations allow for (1) exemplar-based (i.e., count the instances that look similar to the user-provided example), (2) zero-shot (i.e., count the most salient object), and (3) text-prompted counting (i.e., count the text-specified object). Designed to help count high-density images, where continuous-scale predictions are appropriate, they exhibit limited performance in low-density images (Zhang et al., 2025), as illustrated in Fig. 2b. On the other hand, our *D2D* critic is derived from detectors which show robust performance given low-density images, which is critical to the generative setting (Fig. 2). Furthermore, our critic can be used to generate objects in the open set by leveraging *open-vocabulary* detectors, like OWLv2 (Minderer et al., 2023), with minimal modification to detector architecture. In our work, we compare our critic against three regression-based counting methods: RCC (Hobley & Prisacariu, 2022) (zero-shot), CLIP-Count (Jiang et al., 2023) (text-specified), and CounTR (zero-shot) (Chang et al., 2022).

## 3 THE *D2D* FRAMEWORK

**Problem statement.** Given a pre-trained, one-step T2I model $G_\theta$ and prompt $p$ requesting $N$ counts of an object of class $C$, our goal is to generate an image with exactly $N$ counts of $C$.

**Summary of approach.** We propose a detector-based count critic that provides a more effective gradient signal. We then design a method to use that signal to influence the generation process, by leveraging the structural priors in the initial latent which we modify to align with the gradient.

### 3.1 DETECTOR-TO-DIFFERENTIABLE CRITIC

Detector $\mathcal{D}$ takes as inputs an object class $C$ and image $I$ and outputs a set of $n$ bboxes $\{B_i | 1 \le i \le n\}$ and logits $\mathbf{z} = \{z_i | 1 \le i \le n\}$. A standard sigmoid $\sigma(z_i) = \frac{1}{1+e^{-z_i}}$ converts the logits into confidence scores between 0 and 1, with the most salient bboxes filtered using threshold $\tau$, as follows: $\mathbf{B} = \{B_i | \sigma(z_i) \ge \tau\} = \{B_i | z_i \ge \tau_z\}$, where $\tau_z = \sigma^{-1}(\tau)$. The final count is $|\mathbf{B}|$. Our goal is to derive a gradient from $\mathcal{D}$ that can effectively increase or decrease $|\mathbf{B}|$ as needed. Our approach is to first, define a differentiable function $f : \mathbf{z} \in \mathbb{R}^n \mapsto \mathbb{N}$ that can extract the count from the logits $\mathbf{z}$, and second, transform $f$ so its gradient is more amenable to convergence, arriving at critic $\mathcal{L}_{\text{D2D}}$.

**Extract the count via $f$.** The main challenge behind counting is its discrete nature, featuring discontinuous jumps between one count and the next. But converting discrete problems into continuous, differentiable ones has been done before (e.g., logistic regression for binary classification). The task of discrete 0/1 prediction is accomplished by optimizing the steepness and transition threshold of the sigmoid-curve that best splits the classes. By drawing parallels to this space, we arrive at the following insight: we can convert each logit into a binary indicator of whether to "count" the corresponding bbox, by applying to each logit a steep sigmoid curve with transition threshold $\tau_z$ and steepness coefficient $\beta$ (Eq. 1).

**Transform $f$ to effectively handle over/under-generation.** An effective critic provides a strong gradient signal above/below $\tau_z$ (our domain of interest) to push logits beyond or below the threshold as needed to add/erase objects in response to under/over-generation. However, by nature of its sigmoidal shape, $f$ has significant plateauing (i.e., weak gradient signals) above and below $\tau$. To improve the gradient steepness in our domain of interest, we scale each sigmoid output by the corresponding logit (Eq. 2), arriving at $\mathcal{L}_{\text{D2D}}$. At inference-time, we use $\nabla \mathcal{L}_{\text{D2D}}$ to optimize the generated image.[1]

$$f_{\beta,\tau_z}(\mathbf{z}) = \sum_{i=1}^{n} \sigma(\beta \cdot (z_i - \tau_z)). \tag{1}$$

$$\mathcal{L}_{\text{D2D}} = \begin{cases} \sum_{i=1}^{n} \sigma(\beta \cdot (z_i - \tau_z)) \cdot (z_i - \tau_z), & \text{if } f_{\beta,\tau_z} > N \text{ (i.e., over-generation)} \\ \sum_{i=1}^{n} -\sigma(-\beta \cdot (z_i - \tau_z)) \cdot (z_i - \tau_z), & \text{if } f_{\beta,\tau_z} < N \text{ (i.e., under-generation)} \end{cases} \tag{2}$$

**Extension to multiple classes.** The main consideration in extending *D2D* to prompts with $m > 1$ object classes $\{C_j | 1 \le j \le m\}$, is that every bbox comes with $m$ logits, the maximum of which determines its class label. To extend *D2D*, we update Eq. 2 to correct each bbox's largest logit, while minimizing all others. Details in Appendix D.4.

---

[1] Unless otherwise noted, we use $f$ to perform early-stopping once the requested count is met.

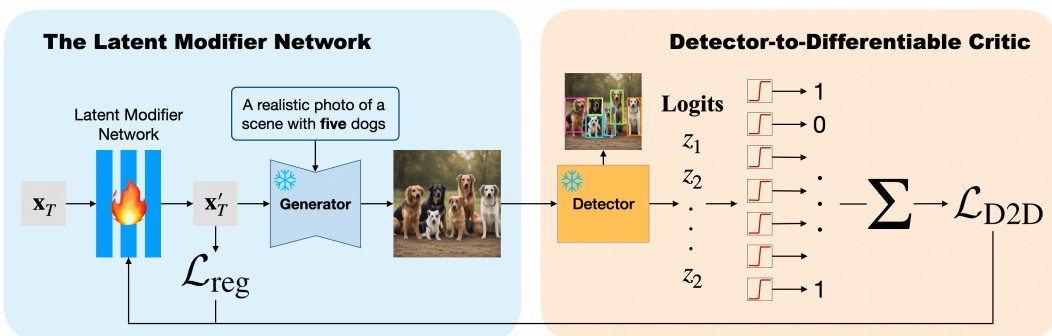

Figure 3: **The *D2D* pipeline for improving T2I numeracy.** *D2D* consists of two main components that work together to improve numeracy: our detector-based count critic guides the Latent Modifier Network (LMN) on how to transform the original initial noise $\mathbf{x}_T$ into a more optimal $\mathbf{x}'_T$. Our count critic uses sigmoid-based activation functions to convert logits into gradient signals, which are then backpropagated through the frozen pipeline to update the weights of the LMN.

### 3.2 THE LATENT MODIFIER NETWORK (LMN)

Given our proposed count critic, we now turn to the learning method used to optimize this objective. Motivated by the presence of meaningful priors in the initial noise, previous works (Eyring et al., 2024; Wang et al., 2025) have used various generic alignment metrics to tune the initial noise directly. Building on this motivation, we propose the Latent Modifier Network (LMN), a test-time tunable module whose output is mixed with the original noise to determine the optimal initial noise and whose weights are updated using our critic $\mathcal{L}_{\text{D2D}}$.

Given initial noise $\mathbf{x}_T \sim \mathcal{N}(0, \boldsymbol{I})$, $\mathbf{x}_T \in \mathbb{R}^d$ and prompt $p$ that requests $N$ counts of an object of class $C$, one-step T2I model $G_\theta$ generates image $I$. Our goal is to find an optimal $\mathbf{x}_T^*$ that produces an image $I^*$ with exactly $N$ of the specified object. To achieve this, we introduce a tunable Latent Modifier Network (LMN) $M_\phi$: a small, 3-layer perceptron, between the initial random latent and T2I model (Fig. 3), with input/output dimensions equal to that of the initial latent and whose output dictates how to update $\mathbf{x}_T$. As shown in Eq. 3, the new latent is a weighted sum of $\mathbf{x}_T$ and $M_\phi(\mathbf{x}_T)$, with weight $w = 0.2$. Compared to tuning the initial latent directly, the LMN composes a relatively larger parameter space and enforces more incremental updates that preserve a consistent Gaussian component sourced from the original latent even through all iterations. At inference-time, we tune $\phi$ using $\nabla\mathcal{L}_{\text{D2D}}$ with the goal of correcting the initial noise, and thereby the numeracy, as described in the following section.

$$\mathbf{x}'_T = w \cdot \mathbf{x}_T + (1 - w) \cdot M_\phi(\mathbf{x}_T). \tag{3}$$

### 3.3 OPTIMIZATION

The goal is to find the optimal set of parameters $\phi$ that minimizes the error between the generated and requested count, as seen in Eq. 4. Since detector $\mathcal{D}$ is non-differentiable, we leverage $\mathcal{L}_{\text{D2D}}$ to optimize $\phi$ iteratively, rendering our final update rule (Eq. 5), with regularization term $\mathcal{L}_{\text{reg}}$, learning rate $\eta$, and weights $\alpha$ and $\lambda$. We adaptively rescale the loss to address exploding gradients that we may encounter due to the large number of tunable parameters. We apply a variant of the regularization term used in ReNO (Eyring et al., 2024), using the negative log-likelihood of the norm of $\mathbf{x}_T$ as follows: $\mathcal{L}'_{\text{reg}} = ||\mathbf{x}'_T||^2/2 - (d-1) \cdot \log(||\mathbf{x}'_T||)$. We use $\mathcal{L}_{\text{reg}} = [a\mathcal{L}'_{\text{reg}} + c]^{10}$, with scaling coefficient $a$ and shift constant $c$.

$$\phi^* = \arg\min_{\phi} |\mathcal{D}(G_\theta(\mathbf{x}'_T)) - N|. \tag{4}$$

$$\phi \Leftarrow \phi - \eta\nabla(\alpha\mathcal{L}_{\text{D2D}} + \lambda\mathcal{L}_{\text{reg}}). \tag{5}$$

$\phi$ **initialization.** To give $M_\phi$ a good starting point (i.e., initialize the network's initial output distribution to Gaussian), we propose a short, pre-inference alignment stage to be performed one time per base model using only the regularization term. Specifically, we train $M_\phi$ on 100 different randomly sampled latents ($\mathbf{x}_T$) for 200 epochs each (Algorithm 2 in the appendix).

At inference-time, given a new, randomly sampled $\mathbf{x}_T$ the network has never seen before, we introduce a ~0.2-second calibration phase to allow the network to adapt to the new input, using only

the regularization term. Afterward, we leverage both *D2D* and regularization terms, according to Eq. 5. The full algorithm is detailed below (Algorithm 1).

---

**Algorithm 1** Inference

---

**Input:** Prompt $p$ specifying $N$ of object of class $C$; pre-trained Latent Modifier Network $M_\phi$; latent dimension $d$; weight $w$, diffusion model $G_\theta$; minimum number of calibration iterations $t_{min}$; threshold value specifying "good enough" regularization $\tau_{reg}$; counter $f_{\beta,\tau_z}$ and critic $\mathcal{L}_{D2D}$; Stage 1 (Calibration) learning rate $\eta_{calib}$ and loss weight $\lambda_{calib}$; Stage 2 numeracy optimization learning rate $\eta$ and loss weights $\alpha$ and $\lambda$; number of tuning steps $K$.

**Output:** Optimal noise $\mathbf{x}_T^*$.

> resample $\leftarrow$ True          ▷ **Stage 1:** Calibrate $M_\phi$ to newly sampled $\mathbf{x}_T$.
> **while** resample **do**
>      Sample $\mathbf{x}_T \in \mathbb{R}^d \sim \mathcal{N}(0, \boldsymbol{I})$
>      **for** $1 \le t \le K$ **do**
>          $\mathbf{x}_T' = w \cdot \mathbf{x}_T + (1 - w) \cdot M_\phi(\mathbf{x}_T)$
>          Compute $\mathcal{L} = \lambda_{calib} \mathcal{L}'_{reg}$
>          **if** $t \ge t_{min}$ and $\mathcal{L} <= \tau_{reg}$ **then**      ▷ Done aligning in $t$ iterations.
>              resample $\leftarrow$ False
>              **break**
>          **else**
>              $\phi \leftarrow \phi - \eta_{calib} \nabla \mathcal{L}$
>          **end if**
>      **end for**
> **end while**
>
> **for** $t \le$ epoch $\le K$ **do**          ▷ **Stage 2:** Optimize numeracy.
>      Compute $\mathcal{L}_{reg}$
>      $I = G_\theta(\mathbf{x}_T', p)$
>      Compute $f_{\beta,\tau_z}$ and $\mathcal{L}_{D2D}$
>      **return** if $f_{\beta,\tau_z} = N$          ▷ if $I$ is optimal, stop early
>      $\phi \Leftarrow \phi - \eta \nabla(\alpha \cdot \mathcal{L}_{D2D} + \lambda \cdot \mathcal{L}_{reg})$
>      $\mathbf{x}_T' = w \cdot \mathbf{x}_T + (1 - w) \cdot M_\phi(\mathbf{x}_T)$
> **end for**

---

# 4 EXPERIMENTS AND ANALYSIS

## 4.1 EXPERIMENTAL SETUP

**Benchmarks.** Our main experimental setting of single-object, low-density prompts leverages two benchmarks, CoCoCount (Binyamin et al., 2025) and D2D-Small. D2D-Small is a set of 400 prompts created using 40 countable objects from COCO (Lin et al., 2014) with counts ranging from 1-10 and a prompt template adapted from Lian et al. (2024): "A realistic photo of a scene with {count} {object}." CoCoCount consists of 200 prompts from 20 COCO classes and requested counts roughly equally split among 2, 3, 4, 5, 7, and 10. Experiments on multi-object or high-density prompts were performed on D2D-Multi (400 prompts with two objects sampled from 40 countable COCO classes, with $N_1, N_2 < 10$) and D2D-Large (similarly constructed with counts 11-20).

**Base models.** We apply *D2D* to three one-step models: SDXL-Turbo (Sauer et al., 2025), SD-Turbo (Sauer et al., 2025), and Pixart-DMD (Chen et al., 2025b). SDXL-Turbo and SD-Turbo, respectively distilled from SDXL (Podell et al., 2024) and SD2.1 (Rombach et al., 2022), have U-Net backbones. Pixart-DMD, distilled from Pixart-$\alpha$ (Chen et al., 2024), has a Transformer backbone.

**Comparison of numeracy enhancement methods.** We compare *D2D* against multi-step count-correction baselines Make It Count (Binyamin et al., 2025), an SDXL-based method which uses attention-based mechanisms to identify and correct object layout via updates to the intermediate latents, and Counting Guidance (Kang et al., 2025), an SD1.4-based method which uses the auxiliary counting RCC (Hobley & Prisacariu, 2022) to correct the predicted noises, and one-step method TokenOpt (Zafar et al., 2024), an SDXL-Turbo-based method that injects a count token into the prompt and tunes it using CLIP-Count (Jiang et al., 2023). Importantly, Make It Count addresses the low-

density, single-object setting and TokenOpt addresses the single-object setting, so we only evaluate Make It Count on CoCoCount and D2D-Small and TokenOpt on CoCoCount and D2D-Small/Large.

**Comparison with generic prompt-alignment method.** The most relevant prior initial noise optimization work is ReNO (Eyring et al., 2024), a framework for one-step T2I models that uses the combined gradient of multiple image quality and prompt-image alignment metrics (ImageReward (Xu et al., 2023), PickScore (Kirstain et al., 2023), HPSv2 (Wu et al., 2023), and CLIPScore (Hessel et al., 2021)) to optimize semantic alignment and image quality. Instead of tuning an LMN, ReNO directly tunes the initial latent over 20-50 iterations, with regularization to keep the noise within the initial distribution and gradient clipping to prevent gradient explosion. Though its use of human-preference reward models does improve numeracy relative to the base model, there remains a gap between using such generic objectives and our count-tailored critic (Tab. 1). A key difference between our method and ReNO's is our introduction of the LMN, which expands the tunable parameter space while preserving a portion of the original initial noise throughout the optimization process. To assess the impact of introducing the LMN, we compare our initial noise optimization method with ReNO's, controlling for the loss by swapping out ReNO's human-preference models for our *D2D* critic.

**Count critic.** We demonstrate *D2D* on detectors OWLv2 (Minderer et al., 2023) (open-vocabulary, robust) and YOLOv9 (Wang et al., 2024) (high-throughput and trained on COCO (Lin et al., 2014) objects). We expect a small accuracy-cost tradeoff, where OWLv2 enables superior numeracy with greater computational overhead, while YOLOv9 yields slightly lower numeracy but faster inference.

**Evaluation.** Following similar evaluation protocols (Binyamin et al., 2025; Kang et al., 2025; Zafar et al., 2024), we use CountGD (Amini-Naieni et al., 2024), a state-of-the-art counting model built on detector GroundingDINO (Liu et al., 2025), to extract the count of generated objects and compute the proportion of correctly-generated images (see Appendix G for CountGD's counting accuracy compared to other regression/detector-based methods). Like Eyring et al. (2024), we analyze image-quality/prompt alignment with human-preference-trained models (ImageReward (Xu et al., 2023), PickScore (Kirstain et al., 2023), HPSv2 (Wu et al., 2023)), and CLIPScore (Hessel et al., 2021).

**Implementation details.** Our main experiments were completed on an L40, with hyperparameter ablations completed on an A6000. For Make It Count (Binyamin et al., 2025) which requires $> 50$ GB, we used an A100. Our key hyperparameters are the detector threshold $\tau$ and steepness coefficient $\beta$, which we set as 0.2 and 300 (ablations reported). Further details in Appendix D.

## 4.2 NUMERACY IMPROVEMENTS

Tab. 1 shows our main *D2D*-to-baseline comparisons. Baseline evaluations illustrate that though the prompt setting is relatively simple, generating accurate counts remains challenging. On numeracy, *D2D* consistently outperforms baselines across low-density, single-object, multi-object, and high-density prompts, across base models with U-Net and DiT backbones. On SDXL-Turbo, we demonstrate that performance boosts from *D2D* generalize across OWLv2 and YOLOv9 detector backbones (i.e., the detector used to compute $\mathcal{L}_{\text{D2D}}$), with a small accuracy-cost tradeoff as expected (Fig. 6). The robust OWLv2 detector yields higher numeracy with slightly more overhead, while the real-time YOLOv9 detector yields slightly lower (but still high) numeracy with faster inference (in all other experiments, we use the higher-performing OWLv2 backbone unless otherwise noted). Additionally, *D2D* effectively complements baselines, boosting numeracy across all four benchmarks when used in combination with TokenOpt or ReNO (Tab. 6 in appendix). For example, applying *D2D*-optimized seeds to TokenOpt improves numeracy by 13.6% points, relative to TokenOpt's baseline performance (from 35.12% to 48.75%) on CoCoCount.

**Improved numeracy on multi-object/high-density prompts.** *D2D* maintains relative improvement over baselines even in the more challenging multi-object/high-density settings. Nevertheless, the accuracy drop from low-density benchmarks to D2D-Large illustrates the remaining challenge of correctly generating large counts. Upon parsing D2D-Multi results, we see this pattern holds even within multi-object prompts (Tab. 7 in appendix). For example, the accuracy of SDXL-Turbo + *D2D* w/ OWLv2 on D2D-Multi prompts with low total-density ($N_{\text{tot}} = N_1 + N_2 \le 10$) is 12.08%, which drops to 3% for prompts with higher $N_{\text{tot}}$ (though both are still higher than all baseline scores).

$\mathcal{L}_{\text{D2D}}$ **effectively boosts numeracy across all classes.** Fig. 4 shows $\mathcal{L}_{\text{D2D}}$ improves numeracy across all 41 object categories in CoCoCount and D2D-Small, spanning a large variety (e.g., apples, elephants, cars, etc.) Upon applying *D2D* to SDXL-Turbo, umbrella and vase are the two classes

Table 1: **Quantitative results.** *D2D* outperforms all baselines across all four benchmarks, even generalizing across detector variants OWLv2 (Minderer et al., 2023) and YOLOv9 (Wang et al., 2024). *D2D* with YOLOv9 *italicized* and **bolded** to show that while it outperforms all baselines, it is second to using OWLv2. Standard deviations indicate the significance of our findings. Base models with no post-enhancement highlighted in gray. Avg. over four seeds.

| Method | CoCoCount | D2D-Small | D2D-Multi | D2D-Large |
|---|---|---|---|---|
| SDXL (Podell et al., 2024) | 24.88 ±1.70 | 16.06 ±1.86 | 2.44 ±0.59 | 1.44 ±0.38 |
| + Make It Count (Binyamin et al., 2025) | 46.75 ±2.10 | 30.00 ±1.93 | —— | —— |
| SDXL-Turbo (Sauer et al., 2025) | 27.38 ±2.69 | 20.31 ±1.95 | 2.12 ±0.83 | 2.56 ±0.55 |
| + ReNO (Eyring et al., 2024) | 41.88 ±1.03 | 27.50 ±0.68 | 5.31 ±0.38 | 4.69 ±1.25 |
| + TokenOpt (Zafar et al., 2024) | 35.12 ±0.75 | 23.31 ±1.66 | —— | 3.94 ±0.72 |
| + *D2D* w/ OWLv2 (Ours) | **55.62** ±2.72 | **43.69** ±2.36 | **9.81** ±0.97 | **9.94** ±1.57 |
| + *D2D* w/ YOLOv9 (Ours) | *52.75* ±1.55 | *36.69* ±2.40 | *6.25* ±1.77 | *7.50* ±1.06 |
| SD2.1 (Rombach et al., 2022) | 32.75 ±1.32 | 24.75 ±2.85 | 4.81 ±1.23 | 2.94 ±0.75 |
| SD1.4 (Rombach et al., 2022) | 27.62 ±4.11 | 16.69 ±2.59 | 2.81 ±0.31 | 2.12 ±0.32 |
| + Counting Guidance (Kang et al., 2025) | 25.25 ±3.75 | 17.12 ±1.69 | 3.38 ±1.16 | 1.88 ±0.60 |
| SD-Turbo (Rombach et al., 2022) | 20.88 ±3.07 | 15.31 ±0.87 | 2.56 ±0.83 | 3.19 ±1.18 |
| + ReNO (Eyring et al., 2024) | 43.38 ±3.47 | 32.06 ±0.99 | 8.94 ±1.76 | 4.25 ±1.14 |
| + *D2D* w/ OWLv2 (Ours) | **48.38** ±3.09 | **39.44** ±2.37 | **10.75** ±1.06 | **11.44** ±1.98 |
| Pixart-α (Rombach et al., 2022) | 19.62 ±1.03 | 14.00 ±1.08 | 1.31 ±0.75 | 1.81 ±0.66 |
| Pixart-DMD (Chen et al., 2025b) | 38.12 ±2.32 | 27.88 ±1.51 | 6.25 ±0.46 | 3.19 ±0.62 |
| + ReNO (Eyring et al., 2024) | 44.75 ±1.44 | 37.25 ±1.70 | 9.44 ±0.75 | 4.75 ±0.74 |
| + *D2D* w/ OWLv2 (Ours) | **53.25** ±2.40 | **41.25** ±2.81 | **13.31** ±1.36 | **7.62** ±1.18 |

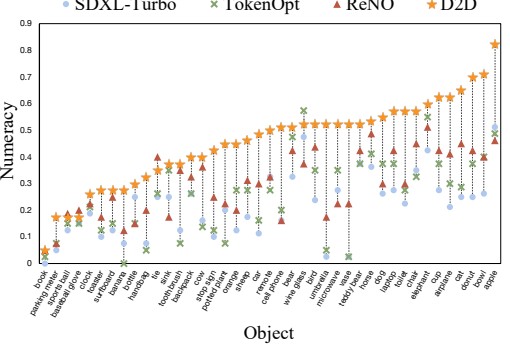

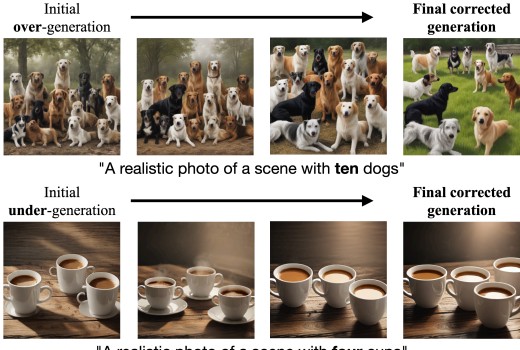

"A realistic photo of a scene with **ten** dogs"

"A realistic photo of a scene with **four** cups"

Figure 4: *D2D* **improves numeracy across all 41 objects in CoCoCount and D2D-Small.** Evaluated against ReNO (Eyring et al., 2024) and TokenOpt (Zafar et al., 2024) on base SDXL-Turbo. Avg. over four seeds.

Figure 5: *D2D* **effectively corrects over and under-generation**. The initial generation contains six more dogs/one fewer cup than requested, which our method iteratively corrects, arriving at an image of 10 dogs/four cups, as requested.

that see the most improvement, each jumping from 3% (base) to 53% (*D2D*) accuracy. Wine glass and bottle, both of which are (semi)transparent objects, are among the classes that see the least improvement (4.8% to 5.3% and 25% to 30% accuracy, respectively), which may suggest a future direction where detectors are fine-tuned on more difficult classes, or similar, with the purpose of generating highly-tailored scenes of objects.

*D2D* **best handles over and under-generation.** Tab. 2 breaks down results by the numeracy of the initial generation $I$, illustrating how well different methods are able to *correct* over/under-generation while *maintaining* the numeracy of already-correct images. Specifically, we compare TokenOpt, ReNO, and *D2D* on base model SDXL-Turbo, across benchmarks CoCoCount and D2D-Small. *D2D* has the highest correction rate, correcting 40.13% of over-generations and 41.83% of under-generations, which is at least 16% points over the baselines, while maintaining minimal decline in

Table 2: **Given the same initial conditions, *D2D* is the most effective at correcting over and under-generation.** We report the correction rate of initial over/under-generations, as well as the proportion of correct generations that were maintained. On SDXL-Turbo, across CoCoCount and D2D-Small benchmarks. Avg. over four seeds.

| Numeracy of initial generation | Over | Under | Correct |
|---|---|---|---|
| TokenOpt (Zafar et al., 2024) | 13.28 | 25.24 | 69.92 |
| ReNO (Eyring et al., 2024) | 23.32 | 25.11 | 62.19 |
| *D2D* w/ OWLv2 | **40.13** | **41.83** | **72.57** |

Table 3: **Ablation study on key hyperparameters $\tau$ and $\beta$.** Detector threshold $\tau = 0.2$ is optimal. A lower $\tau$ (which counts low-confidence bboxes) and higher $\tau$ (which potentially discards actually-legitimate bboxes) results in drops in numeracy. Steepness coefficient $\beta = 300$ is optimal. Tested on CoCoCount, seed=0.

| Hyperparameters | $\tau$ | | | | $\beta$ | | | | |
|---|---|---|---|---|---|---|---|---|---|
| | 0.1 | 0.2 | 0.5 | 0.8 | 1 | 10 | 20 | 300 | 400 |
| CountGD | 51.50 | **55.50** | 43.50 | 32.50 | 43.00 | 40.00 | 32.50 | **55.50** | 52.50 |

Table 4: **Among count critics, $\mathcal{L}_{\text{D2D}}$ is the most effective.** On SDXL-Turbo. Avg. over four seeds.

| Count Critic | CoCoCount | D2D-Small | D2D-Multi | D2D-Large |
|---|---|---|---|---|
| RCC (Hobley & Prisacariu, 2022) | 37.75 | 26.38 | —— | 04.25 |
| CLIP-Count (Jiang et al., 2023) | 40.00 | 25.88 | 05.19 | 06.38 |
| CounTR (Chang et al., 2022) | 38.38 | 25.62 | —— | 05.31 |
| $f$ (OWLv2) | 32.00 | 20.75 | 03.06 | 03.38 |
| $\mathcal{L}_{\text{D2D}}$ (OWLv2) | **55.62** | **43.69** | **09.81** | **09.94** |

correct generations. Fig. 5 illustrates *D2D*'s iterative correction process on two sample prompts, going from 16 dogs to the requested 10 dogs and from three cups to the requested four.

### 4.3 ADDITIONAL ANALYSIS AND ABLATIONS

**Impact of hyperparameters.** We report our hyperparameter studies on values for $\tau$ (detector threshold) and $\beta$ (steepness coefficient). Results (Tab. 3) show that $\tau = 0.2$, $\beta = 300$ are optimal.

***D2D* vs. regression-based counters.** Tab. 4 compares the effectiveness of our critic against existing regression-based ones and additionally shows that the formulation $\mathcal{L}_{\text{D2D}}$ is indeed more convergence-friendly than $f_{\beta,\tau_z}$. Across all four benchmarks, our detector-based critic outperforms regression-based methods RCC, CLIP-Count, and CounTR on numeracy (e.g., ours reaches 55.62% when the max score reached by any regression-based model is 40% on CoCoCount). Notably, $\mathcal{L}_{\text{D2D}}$ outperforms even on the high-density benchmark D2D-Large, though regression-based methods outperform detectors in the non-generative, counting setting (Fig. 2b). Furthermore, not only does $\mathcal{L}_{\text{D2D}}$, which produces a stronger gradient signal, outperform $f_{\beta,\tau_z}$ on numeracy; $f_{\beta,\tau_z}$ yields the lowest numeracy compared to the other critics, which indicates that though it composes the mathematical backbone of $\mathcal{L}_{\text{D2D}}$, $f_{\beta,\tau_z}$ itself is not a suitable critic, as expected (Tab. 4).

**The latent modifier network $M_\phi$.** Next, we assess the impact of introducing the LMN, a module whose output is mixed with the original noise to arrive at the optimal noise, by comparing our method with ReNO's, controlling for the optimization objectives used ($\mathcal{L}_{\text{D2D}}$, $\mathcal{L}'_{\text{reg}}$) and number of iterations tuned. Tab. 5 shows the LMN generally improves numeracy, while maintaining image quality; numeracy jumps 10% points on CoCoCount and D2D-Small from 43.25% to 53.88% and from 32% to 42.44%, respectively.

**Impact on image quality and computational overhead.** ImageReward, PickScore, HPSv2, and CLIPScore metrics in Fig. 6a show *D2D*'s image quality and overall prompt alignment is comparable to counting baselines and even surpasses multi-step baselines in many cases, including the layout control-based method, Make It Count (MIC). For example, SDXL-Turbo + *D2D* (OWLv2) yields ImageReward 0.51 (MIC: 0.30), PickScore 21.98 (MIC: 21.48), and HPSv2 0.28 (MIC: 0.26) on D2D-Small. *D2D* does not add significantly to inference cost, averaging between 11 and 21 seconds, compared to counting baselines, which average upwards of 28 to 100 secs (Fig. 6b).

## 5 CONCLUSION AND DISCUSSIONS

In this work, we address the challenge of correcting numeracy in generation. We identify a central limitation of previous methods, specifically their reliance on differentiable, regression-based counting models as critics. We propose a novel way to convert more robust detectors into differentiable count critics and then use them to optimize the initial noise at inference-time to improve numeracy. Our

Table 5: **The LMN boosts numeracy.** We compare *D2D* against ReNO (Eyring et al., 2024) using $\mathcal{L}_{\text{D2D}}$ and $\mathcal{L}'_{\text{reg}}$ for both, controlling for the number of iterations tuned. We note boosts in numeracy, with comparable image quality. On SDXL-Turbo. Avg. over four seeds.

| Method | CountGD ↑ | | ImageReward ↑ | | PickScore ↑ | | HPSv2 ↑ | | CLIPScore ↑ | |
|---|---|---|---|---|---|---|---|---|---|---|
| | CoCoCount | D2D-Small | CoCoCount | D2D-Small | CoCoCount | D2D-Small | CoCoCount | D2D-Small | CoCoCount | D2D-Small |
| ReNO w/ $\mathcal{L}_{\text{D2D}}$, $\mathcal{L}'_{\text{reg}}$ | 43.25 | 32.00 | 1.04 | 0.45 | 23.25 | 21.98 | 0.296 | 0.281 | **32.81** | **31.79** |
| *D2D* w/ $\mathcal{L}_{\text{D2D}}$, $\mathcal{L}'_{\text{reg}}$ | **53.88** | **42.44** | **1.08** | **0.52** | **23.28** | **21.99** | **0.299** | **0.282** | 32.77 | 31.71 |

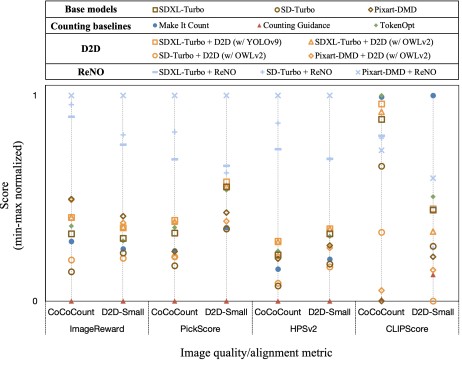 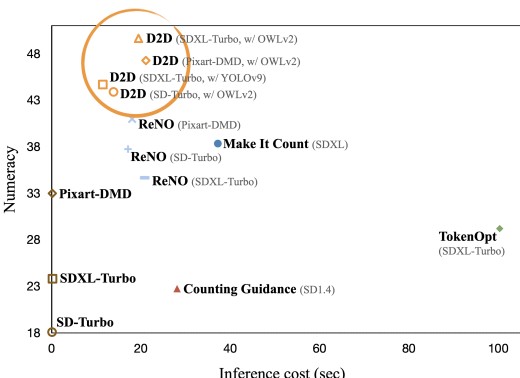

(a) Image quality/alignment scores (ImageReward (Xu et al., 2023), PickScore (Kirstain et al., 2023), HPSv2 (Wu et al., 2023), CLIPScore (Hessel et al., 2021)) by method. Aside from ReNO (Eyring et al., 2024), which often scores highest (it specifically optimizes those metrics), *D2D* is comparable to counting baselines. Min-max normalized.

(b) Numeracy vs. inference cost by method. Across base models (SDXL-Turbo, SD-Turbo, Pixart-DMD) and detectors (OWLv2 (Minderer et al., 2023), YOLOv9 (Wang et al., 2024)), *D2D* scores in the top left (i.e. it is both high-numeracy and low-cost). *D2D* w/ YOLOv9 is even more compute-efficient than w/ OWLv2. Base model/detector noted in gray.

Figure 6: ***D2D* yields image quality/alignment comparable to counting baselines, with minimal addition to computational overhead.** Comparisons against counting baselines (Make It Count (Binyamin et al., 2025), Counting Guidance (Kang et al., 2025), TokenOpt (Zafar et al., 2024)) and generic alignment method ReNO. On CoCoCount and D2D-Small. Avg. over four seeds.

method yields the highest numeracy across various prompt scenarios, including low-density, single-object, multi-object, high-density settings, effectively correcting both over and under-generation, with minimal additions to temporal overhead and minimal degradation in image quality.

**Limitation and future directions.** While our method exhibits significant improvements in numeracy, high-density scenarios remain challenging. Given regression-based methods are more appropriate in this setting, a future direction may explore how to adapt them into the generative setting. *D2D* is limited in more fine-grained control (e.g., object placement) as it avoids direct enforcement and layout control, which can come at the cost of image quality. Furthermore, *D2D* is inherently bottlenecked by detector performance, though detectors are relatively robust. Future directions may explore using *D2D* to perform other complex tasks, like attribute binding and object positioning, leveraging detectors that can robustly work with prompts specifying objects and associated attributes.

### REPRODUCIBILITY STATEMENT

The paper, appendix, along with code that we will release, contain the details for reproducibility.

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
