# OpenReview forum: "D2D: Detector-to-Differentiable Critic for Improved Numeracy in Text-to-Image Generation"
_ICLR.cc/2026/Conference — Submitted to ICLR 2026_

### Official Review · Reviewer_Zbh8 · 2025-10-24

**Soundness:** 2
**Presentation:** 2
**Contribution:** 2
**Rating:** 4
**Confidence:** 3

**Summary:**

This paper addresses the challenge of diffusion models failing to generate the precise number of objects specified in a prompt. While many existing methods employ gradients of regression-based counting networks, the authors argue that detection models, such as YOLO, offer superior count accuracy. To overcome the non-differentiable nature of these detectors, they introduce a novel surrogate loss function. This loss enables optimization by updating the initial noise, a technique similar to that in Reno, rather than relying on traditional gradient guidance during the diffusion process.

**Strengths:**

- The work demonstrates strong quantitative and qualitative performance against baselines, validated across multiple benchmarks and models, while also achieving low inference latency

**Weaknesses:**

- The novelty of this work appears limited, primarily consisting of replacing regression-based counting networks with detection models wrapped by a differentiable loss. Other parts of the methodology seem largely similar to previous work.
- Including Algorithm 2 from the appendix in the main paper would significantly aid in understanding the proposed algorithm.
- An explanation and analysis on why detection models are superior to regression models for this application would be very helpful and strengthen the paper's claims.
- Minor: The paper contains some grammar mistakes and typos that should be corrected throughout.

**Questions:**

- Inconsistency regarding High Object Counts: Figure 2b indicates that regression models begin to outperform detection models at higher object counts. However, Table 1 shows that your approach still outperforms the baselines in these same high-count scenarios. Could the authors please explain this apparent discrepancy?
- Computational Cost of $L_{\text{D2D}}$: Examining Algorithm 2, it appears that computing the loss term $L_{\text{D2D}}$ requires backpropagating through all diffusion steps. Could the authors elaborate on why this only has a minor effect on latency? Furthermore, what are the implications of this on memory usage during training and inference?
- Handling Duplicate Bounding Boxes: Detection models like YOLO typically utilize post-processing steps, such as Non-Maximum Suppression (NMS), to eliminate duplicate bounding boxes. How does your method handle the issue of potentially overlapping or duplicate detections?

---

> ### Author Response · Authors · 2025-11-21
>
> >W1: The novelty of this work appears limited, primarily consisting of replacing regression-based counting networks with detection models wrapped by a differentiable loss. Other parts of the methodology seem largely similar to previous work.
>
> **A1:** Thank you for raising this question. We agree that our core contribution is in allowing T2I models to leverage detectors as differentiable count critics at inference time, but we would like to highlight the following:
>
> - Numeracy in DMs vs. larger models: We performed additional evaluations on larger and newer models and observed that numeracy limitations persist (R-LUcL's W1 and our response A1). We believe this underscores an intrinsic challenge in generative models, whose **distributional learning tends to lack discriminative abilities required for accurate numeracy**. Integrating improved discriminative capability into modern generative models is a fundamental challenge, with numeracy being one representative anchor. Furthermore, our empirical results demonstrate that by integrating the proposed D2D with smaller models like SDXL-Turbo, we can already achieve better numeracy accuracy than larger models such as SD3.5 Large Turbo.
>
> - D2D solves a major bottleneck in existing numeracy optimization frameworks. Previous gradient-guidance based works like Counting Guidance [a] and TokenOpt [b] were limited to less accurate, regression-based counters, because detectors are not immediately differentiable. Furthermore, even if discrete counts can be converted into "soft" differentiable versions, making it also convergence-friendly is non-trivial, as sigmoids by nature have weak gradient signals in most of their domain. As empirically demonstrated in row 4 of Table 4 (lines 444-450), which compares the various count critics, the simple differentiable count surrogate $f$ (OWLv2) performs even worse than regression-based counters, but applying our convergence-friendly scaling transforms the loss into the most effective critic (row 5, Table 4).
>
> - Numeracy optimization requires two key components: **(1)** an accurate understanding of the current generated count, which is crucial to early stopping correctly and **(2)** a strong gradient for effective correction as needed (adding/deleting objects). Previous works like TokenOpt rely on two separate counting models to provide these two key capabilities, specifically because detectors are good counters but are not differentiable while regression-based models have gradients but are not good counters. For example, TokenOpt uses CLIP-Count [c] for its gradient signal and YOLO [d] to provide accurate count understanding, thereby increasing the inference cost by 2-6 times, compared to D2D, since the same image is processed through both models. On the other hand, D2D offers a better tradeoff between numeracy and inference time, because we formulate our approach to extract two important things from the same detector: (1) the count and (2) a convergence-friendly gradient.
>
> >W2: Including Algorithm 2 from the appendix in the main paper would significantly aid in understanding the proposed algorithm.
>
> **A2:** Thank you for the suggestion. We have updated the manuscript and added Algorithm 2 to the main text on lines 273-300 (now labeled Algorithm 1).
>
> >W3: An explanation and analysis on why detection models are superior to regression models for this application would be very helpful and strengthen the paper's claims.
>
> **A3:** Thank you for the insightful question. Based on our analysis of the baseline counting accuracies of detectors compared to regression-based models (Fig. 2 on lines 108-124), we observe that detectors are more robust, as also observed in previous works [e,f]. Furthermore, this performance boost generalizes to the generative setting, because accurate count understanding is one of the key components of numeracy optimization. As for the question of *why* detectors are more robust in the first place, we refer to existing work [f], which indicates this may be due to detectors' more advanced localization ability. In the low-density setting, objects are more likely to be sparsely separated, so it becomes essential to localize each instance in order to obtain a count; detectors count explicitly by localizing instances first, whereas regression-based methods capture the density of the crowd of instances, generally without specific localization objectives. In the high-density setting, when object instances are too small and potentially occluded, localization may become more difficult, thereby reducing the accuracy of detector counts.

---

> ### Author Response · Authors · 2025-11-21
> **Rebuttal by Authors**
>
> >W4: Minor: The paper contains some grammar mistakes and typos that should be corrected throughout.
>
> **A4:** Thank you for the careful reading, we have thoroughly checked the manuscript and made corrections in the revised version.
>
> >Q1: Inconsistency regarding High Object Counts: Figure 2b indicates that regression models begin to outperform detection models at higher object counts. However, Table 1 shows that your approach still outperforms the baselines in these same high-count scenarios. Could the authors please explain this apparent discrepancy?
>
> **A5:** Thank you for the keen observation. We believe this may be due to an inherent difference in target objective between the counting task vs. generative task. In counting (i.e. passive understanding), the image is fixed and the goal is to accurately **capture** the number of already-visible instances. In generation (i.e. active optimization and editing of the image), the requested count is fixed and the goal is to **modify** the image to show the correct number of visible object instances. Despite that regression-based methods outperform detectors in the high-density count-capturing task, we believe that *in the generative task* detectors may bring an added benefit in actively updating the image to best suit the kinds of count scenarios it is best at understanding (i.e. scenarios where object instances are well-separated and large enough to localize accurately [f]).
>
> >Q2: Computational Cost of $L_{\text{D2D}}$: Examining Algorithm 2, it appears that computing the loss term $L_{\text{D2D}}$ requires backpropagating through all diffusion steps. Could the authors elaborate on why this only has a minor effect on latency? Furthermore, what are the implications of this on memory usage during training and inference?
>
> **A6:** Thank you for bringing up this key point. D2D works with one-step models (SDXL-Turbo, SD-Turbo, Pixart-DMD), so the gradient only needs to be backpropagated through one diffusion step each time. For clearer comparison of latency, we have replaced Table 6 (previously containing the inference cost column) with a visual plot (Fig. 6b on lines 494-515). While the latency does increase relative to base models, the numeracy-latency tradeoff is best, compared to counting baselines, because D2D uses robust, detector-based early stopping (when the requested count is met). D2D on SDXL-Turbo, for example, requires around 28 GB of GPU VRAM.
>
> >Q3: Handling Duplicate Bounding Boxes: Detection models like YOLO typically utilize post-processing steps, such as Non-Maximum Suppression (NMS), to eliminate duplicate bounding boxes. How does your method handle the issue of potentially overlapping or duplicate detections?
>
> **A7:** Thank you for this question. Our method handles this by introducing the sensitivity hyperparameter $\tau$, which is interpreted as the confidence threshold above which bounding boxes will be "counted." A higher $\tau$ would filter out low-confidence bounding boxes (potentially overlapping/duplicate ones). We find $\tau=0.2$ is optimal for our setting, as shown in Table 3 (lines 439-442).
>
> ---
> [a]  Counting guidance for high fidelity text-to-image synthesis. WACV '25.
>
> [b] Iterative object count optimization for text-to-image diffusion models. arXiv.
>
> [c] Clip-count: Towards text-guided zero-shot object counting. ACMMM '23.
>
> [d] YOLO-v1 to YOLO-v8, the rise of YOLO and its complementary nature toward digital manufacturing and industrial defect detection. Machines. 2023.
>
> [e] Improving CLIP Counting Accuracy via Parameter-Efficient Fine-Tuning. TMLR. 2025.
>
> [f] DecideNet: Counting varying density crowds through attention guided detection and density estimation. CVPR '18.

---

> ### Comment · Reviewer_Zbh8 · 2025-11-28
> **Reply to Authors**
>
> Thank you for the detailed response and for updating the manuscript with the algorithm in the main text. Most of my concerns were addressed, particularly regarding the handling of duplicate bounding boxes and the computational clarifications.
>
> However, the paper seems to have two crucial limitations which hinder me from raising my score:
>
> 1.  **Limited Novelty:** I still believe that replacing regression counting models with detection-based counting models is an insufficient contribution. The paper builds on the fact that current detection models perform better than regression models, but I believe this is largely because detection models are currently more popular, practical, and explored than the latter. This performance gap is not necessarily fundamental and may not persist in the future, which weakens the long-term impact of the proposed shift.
> 2.  **Restricted Scope (Single-step models):** As confirmed in your response (A6), the method positions itself as only applying to single-step diffusion models (e.g., SDXL-Turbo) to maintain latency. This is a significant restriction compared to other baselines, such as Counting Guidance [a], which are applicable across a broad range of diffusion models.
>
> Given these remaining limitations regarding the scope and the core contribution, I will maintain my current score.

---

> ### Author Response · Authors · 2025-12-03
> **Rebuttal by Authors**
>
> Thank you for the additional feedback. We appreciate the opportunity to clarify the remaining two points.
>
> >W1: Limited Novelty: I still believe that replacing regression counting models with detection-based counting models is an insufficient contribution. The paper builds on the fact that current detection models perform better than regression models, but I believe this is largely because detection models are currently more popular, practical, and explored than the latter. This performance gap is not necessarily fundamental and may not persist in the future, which weakens the long-term impact of the proposed shift.
>
> **A8:** Novelty: D2D solves a fundamental limitation, not a temporary performance gap.
>
> (a) **Prior work explicitly declared detector-based guidance impossible.** For example, TokenOpt [b] notes "While detection-based methods offer strong cues for object localization and counting, they rely on non-differentiable thresholding operations, which makes them unsuitable for direct gradient-based optimization." As shown, this was considered a fundamental limitation of inference-time numeracy optimization.
>
> (b) **The novelty lies in enabling a previously impossible class of critics.** Our contribution is not just "swapping out the regression-based module with a detector, because it works better." In fact, we show that simple swapping does not work. Table 4 on lines 444 to 450 (also included below) shows that a simple sigmoid-based, detector-to-counter conversion (which is what $f$ does), performs even worse than regression-based methods. So, we see that our core contribution is not just in this simple swapping.
>
> Rather, we show that it is necessary to transform $f$ into a more convergence friendly form to see the tangible benefits of the detector-based critic. Our design of the critic loss, along with Latent Modifier Network, is our core contribution. We conducted experiments with alternative critic formulations, as demonstrated in the second table below, and show that our proposed critic formulation is best, both in terms of numeracy and inference time.
>
> Table 4 from paper. On SDXL-Turbo. Avg. over four seeds.
> | Count Critic | CoCoCount | D2D-Small | D2D-Multi | D2D-Large |
> |---------|---------|---------|---------|---------|
> | RCC | 37.75 | 26.38 | — | 04.25 |
> | CLIP-Count | 40.00 | 25.88 | 05.19|  06.38 |
> | CounTR |  38.38 | 25.62 | — | 05.31 |
> | $f$ (OWLv2) | 32.00 | 20.75 | 03.06 | 03.38 |
> | $L_{\text{D2D}}$ (OWLv2) | **55.62** | **43.69** | **09.81** | **09.94** |

---

> > ### Author Response · Authors · 2025-12-03
> > **Rebuttal by Authors**
> >
> > Alternative critic formulations. On CoCoCount, seed=0 on SDXL-Turbo.
> > | Scaling formulation                                      | Numeracy | Inference time (s) |
> > |--------------------------------------------------------------|--------------|---------------------------|
> > | $\frac{1}{1+e^{-\beta(z-\tau)}} \cdot (z-\tau)$ (ours)  |   **55.5**       |  **17.10**                 |
> > | $e^{z-\tau}$                                                      |    52.0      | 56.28                 |
> > | $e ^ \sigma(\beta(z - \tau))$                             |    31.5       |    32.24              |
> > | $z-\tau$                             |    50.5       |    53.23             |
> > | $ln(1+e^{z-\tau})$                             |    55.0       |    20.09             |
> > | $(z-\tau) \cdot e^{z-\tau}$                             |    49.0       |    72.72             |
> > | $(z-\tau) \cdot ln(1+e^{z-\tau})$                             |    54.0       |    22.31             |
> > | $\sigma(\beta(z-\tau)) \cdot e^{z-\tau}$                             |    50.5      |    63.96             |
> > | $\sigma(\beta(z-\tau)) \cdot ln(1+e^{z-\tau})$                             |    55.0       |    18.65    |
> >
> > We believe the numeracy boost is attributed to the intentional design of the loss curvature in domains of interest. The idea is that we want to design a fully-smooth loss that will effectively update each bbox's logit in order to bring the total count closer to the requested count. To illustrate, if the total generated count is bigger than requested count, then the overall objective is to bring the logits down. Practically speaking, on the level of individual logits, this means focusing on bringing down the logits that are above the $\tau$ and allowing some flexibility for logits that are already below $\tau$ so that the optimization can focus on the logits greater than $\tau$. This points to the motivation that we want a strong gradient signal above $\tau$ and little signal below. With this in mind, we added to the appendix Figure 14 on lines 1316-1346 plotting the curves of the first three scaling formulations from the table, illustrating how they behave above and below $\tau$. Our formulation has a strong signal above and weak signal below $\tau$. $e^{z-\tau}$ still maintains a significant gradient below $\tau$. $e^{\frac{1}{1+e^{\beta(z-\tau)}}}$ still shows plateaus above $\tau$. Appendix K (starting on line 1296) includes these details.
> >
> > (c) **From a research motivation perspective, the detector/regression-based gap is fundamental in the generative setting.** The aim of generative numeracy is to produce clean, localized, easily countable instances. And detectors provide exactly the signal needed (explicit, instance-specific, localization-centered), because detection evaluates instance localization and instance-specific model confidence. Our method leverages the correct inductive bias for generation, not just a temporarily stronger model family. Our work also highlights an important opportunity for future regression models—to incorporate more instance-level structure so they better reflect the inductive biases required for generative numeracy.

---

> ### Author Response · Authors · 2025-12-03
> **Rebuttal by Authors**
>
> >W2: Restricted Scope (Single-step models): As confirmed in your response (A6), the method positions itself as only applying to single-step diffusion models (e.g., SDXL-Turbo) to maintain latency. This is a significant restriction compared to other baselines, such as Counting Guidance [a], which are applicable across a broad range of diffusion models.
>
> **A9:** We would like to clarify that D2D is not limited to single-step models in principle, and its practical performance dominates methods with broader focus.
>
> While our main experiments reported in the paper mainly focus on one-step models, our proposed D2D critic can also be applied to multi-step base models as demonstrated in our additional experiments in the rebuttal. To illustrate this is possible, we applied the proposed D2D critic to Direct Noise Optimization [g], an initial noise optimization framework for multi-step models. The results have been added to Appendix M on lines 1374-1395.
>
> We also note that the methodological design of Counting Guidance may have its own generalization limitations. While our method directly optimizes the initial Gaussian noise, making it widely applicable across diffusion backbones (including DiT) with minimal additional tuning of optimization parameters, Counting Guidance updates the predicted noise at each diffusion step, which, as noted in InitNO [h], is an approach that requires careful choice of optimization parameters to ensure the predicted noises at each step are moving toward the optimum but not straying away from the pretrained distribution.
>
> -------------------
> [g] Inference-Time Alignment of Diffusion Models with Direct Noise Optimization. ICML '25.
>
> [h] InitNO: Boosting Text-to-Image Diffusion Models via Initial Noise Optimization. CVPR '24.

---

### Official Review · Reviewer_LUcL · 2025-10-28

**Soundness:** 4
**Presentation:** 3
**Contribution:** 2
**Rating:** 4
**Confidence:** 4

**Summary:**

The paper presents an inference-time tuning strategy for improving the numeracy/counting of text-to-image models. The main idea is to solve an inference-time optimization process for the initial noise to optimize the count obtained from an object detector. From a technical standpoint, the main contributions are in a) creating a differentiable critic by applying a sigmoid activation over the logits b) modulating the initial noise with the "latent modifier network". Results on several counting benchmarks clearly indicate that this inference-time optimization is able to drastically improve the numeracy of these models.

**Strengths:**

The biggest strength of the paper is in constructing an effective objective for differentiable optimization. This ensures that there's effective inference-time optimization, which is visible from the strong empirical results on several counting benchmarks.

The paper is also well-presented and easy to follow.

**Weaknesses:**

Avoids more recent larger models: The results in the paper are on the SD-Turbo, SDXL-Turbo and Pixart-alpha models which are all not only distilled (therefore a bit worse in performance), but also fairly out of date as of late 2025 (all being released between late 2023-early 2024). While some of the newer models may not be applicable, one could still see results on Flux-Schnell, SANA-Sprint, SD3.5-Turbo to see how effective this optimization framework is on newer problems. While I'd expect the counting problem to still be there for newer/larger models, I'd imagine that it is a lot less of a problem with these models (e.g. Qwen-Image etc.)


Limited Contribution: The paper is essentially applying the existing initial noise optimization formulation for improved numeracy in T2I models (which had in the past either been used for toy objectives or general human preference/prompt alignment). While effective, from a conceptual standpoint it does feel somewhat limiting, especially since it inherits the issues of inference-time optimization (i.e slower runtimes, memory overhead etc.). To that extent, while the paper's contribution itself might be an integral part of a modern diffusion models' post-training pipeline (since it's easy to verify similar to the RL formulation in LLMs for math/code), it feels rather minimal to merit acceptance in my view.

**Questions:**

In general, I'm mostly curious about how relevant this problem is to be tackled on a standalone basis with a somewhat expensive inference-time optimization formulation. In that regard, a) showing how much of an issue this is on recent, larger models, b) ability of the noise optimization formulation to work with at least the newer distilled models, c) whether these techniques could be used to enhance the base capabilities of this model (i.e fine-tuning a base model on a high-quality dataset of optimized samples etc.) would go a long way in strengthening the contributions of the paper. While I don't really have major doubts about the "soundness" or "presentation" of the paper, I would really like to see more evidence about the "contributions" of the paper before recommending acceptance for the paper.

---

> ### Author Response · Authors · 2025-11-21
> **Rebuttal by Authors**
>
> >W1: Avoids more recent larger models: The results in the paper are on the SD-Turbo, SDXL-Turbo and Pixart-alpha models which are all not only distilled (therefore a bit worse in performance), but also fairly out of date as of late 2025 (all being released between late 2023-early 2024). While some of the newer models may not be applicable, one could still see results on Flux-Schnell, SANA-Sprint, SD3.5-Turbo to see how effective this optimization framework is on newer problems. While I'd expect the counting problem to still be there for newer/larger models, I'd imagine that it is a lot less of a problem with these models (e.g. Qwen-Image etc.)
>
> **A1:** We appreciate the thoughtful comment and have conducted additional experiments on newer models as suggested (table below). **Counting remains a problem:** even newer/larger models like Flux-Schnell still struggle with accurate counting (46.0% numeracy), confirming the continued relevance of this problem. We have added this table to Appendix O on lines 1421-1432.
>
> | Model | Numeracy|
> |-------------------------------|-----------------|
> |FLUX.1 [schnell] (1-step) |  46.0  |
> |FLUX.1 [schnell] (4 steps) |  47.5  |
> | SD3.5 Large Turbo (4-steps) |  48.0   |
> | SDXL-Turbo                     |  30.0      |
> | SDXL-Turbo + D2D            |  **55.5**   |
>
>
> >W2: Limited Contribution: The paper is essentially applying the existing initial noise optimization formulation for improved numeracy in T2I models (which had in the past either been used for toy objectives or general human preference/prompt alignment). While effective, from a conceptual standpoint it does feel somewhat limiting, especially since it inherits the issues of inference-time optimization (i.e slower runtimes, memory overhead etc.). To that extent, while the paper's contribution itself might be an integral part of a modern diffusion models' post-training pipeline (since it's easy to verify similar to the RL formulation in LLMs for math/code), it feels rather minimal to merit acceptance in my view.
>
> **A2:** Thank you for raising this question. We would like to clarify our contributions with respect to post-training methods and the broader context.
>
> 1. Inference-time optimization vs. fine-tuning: This is an insightful question, echoing Q5 and our response A10 to R-NJkJ. Both inference-time optimization and fine-tuning ultimately aim to steer the base distribution toward a target tilted distribution defined by new post-training objectives. Each approach has trade-offs: inference-time optimization can incur slower runtimes and memory overhead, as noted by the reviewer, while fine-tuning alters the original model weights and limits downstream generalization. In the specific case of numeracy, fine-tuning the entire base distribution for precise counting is often more difficult than steering outputs towards a particular prompt with specific counts and object labels. Our method offers an effective way to enhance numeracy via inference-time optimization.
>
> 2. Numeracy in DMs vs. larger models: As suggested by the reviewer, we evaluated larger and newer models (see A1) and observed that numeracy limitations persist. We believe this underscores an intrinsic challenge in generative models, whose **distributional learning tends to lack discriminative abilities required for accurate numeracy**. Integrating improved discriminative capability into modern generative models is a fundamental challenge, with numeracy being one representative anchor. Furthermore, our empirical results demonstrate that by integrating the proposed D2D with smaller models like SDXL-Turbo, we can already achieve better numeracy accuracy than larger models such as SD3.5 Large Turbo.
>
> 3. Numeracy optimization requires two key components: **(1)** an accurate understanding of the current generated count, which is crucial to early stopping correctly and **(2)** a strong gradient for effective correction as needed (adding/deleting objects). Previous works like TokenOpt rely on two separate counting models to provide these two key capabilities, specifically because detectors are good counters but are not differentiable while regression-based models have gradients but are not good counters. For example, TokenOpt uses CLIP-Count [a] for its gradient signal and YOLO [b] to provide accurate count understanding, thereby increasing the inference cost by 2-6 times, compared to D2D, since the same image is processed through both models. On the other hand, D2D offers a better tradeoff between numeracy and inference time, because we formulate our approach to extract two important things from the same detector: (1) the count and (2) a convergence-friendly gradient.

---

> ### Author Response · Authors · 2025-11-21
> **Rebuttal by Authors**
>
> >Q1: In general, I'm mostly curious about how relevant this problem is to be tackled on a standalone basis with a somewhat expensive inference-time optimization formulation. In that regard, a) showing how much of an issue this is on recent, larger models, b) ability of the noise optimization formulation to work with at least the newer distilled models, c) whether these techniques could be used to enhance the base capabilities of this model (i.e fine-tuning a base model on a high-quality dataset of optimized samples etc.) would go a long way in strengthening the contributions of the paper. While I don't really have major doubts about the "soundness" or "presentation" of the paper, I would really like to see more evidence about the "contributions" of the paper before recommending acceptance for the paper.
>
> **A3:** Thank you for the suggestion. We have added experiments on newer models such as SD3.5 Large Turbo and Flux.1 [schnell]. These results show that even the most recent models still struggle with numeracy, with peak performance around 48.0%. In contrast, applying D2D to a base model like SDXL-Turbo—whose unoptimized numeracy is lower (30.0%)—raises its numeracy to 55.5%, surpassing the newer and larger models.
>
> ---
> [a] Clip-count: Towards text-guided zero-shot object counting. ACM MM '23.
>
> [b] YOLO-v1 to YOLO-v8, the rise of YOLO and its complementary nature toward digital manufacturing and industrial defect detection. Machines. 2023.

---

### Official Review · Reviewer_Jery · 2025-11-03

**Soundness:** 3
**Presentation:** 3
**Contribution:** 3
**Rating:** 6
**Confidence:** 4

**Summary:**

This paper tackles the problem of improving object numeracy in text-to-image diffusion models, which often fail to generate the correct number of objects specified in text prompts.
The paper proposes Detector-to-Differentiable (D2D), a novel framework that converts non-differentiable object detectors into differentiable critics.
By optimizing the initial noise during inference, D2D enhances count accuracy while preserving image quality.
Experiments across multiple T2I backbones and benchmarks show meaningful improvement in object counting accuracy with minimal computational overhead.

**Strengths:**

1. The proposed method is logical and well-motivated, with clear explanations that make the underlying rationale and implementation easy to understand.

2. The analysis of the proposed framework is in-depth: for instance, the comparison between detector-based and regression-based counting models (Figure 2), the exploration of class-wise performance (Figure 4), and the ablation between direct noise optimization and LMN-based methods (Table 5).

3. The demonstrated generality of the proposed method across diverse diffusion backbones and evaluation scenarios is commendable.

**Weaknesses:**

1. The evaluation relies solely on a detector-based protocol to assess the proposed detector-based critic. As noted around line 315, the paper employs the SOTA counting model CountGD (built upon GroundingDINO). Although the proposed critic uses different detectors such as OWL-ViT or YOLO for initial noise optimization, this setup risks giving an unfair advantage aligned with the evaluation criterion. To more robustly validate the superiority of the method, additional evaluation metrics, such as human evaluation (as adopted in Make It Count, CVPR 2025), should be considered.

2. It would be interesting to investigate whether the proposed noise optimization process restricts the diversity of generated images. Since the method optimizes the initial noise, the input noise distribution may become narrower than that of purely random noise, potentially reducing image diversity. Thus, analyzing generation diversity would provide important insights into possible trade-offs.

3. Table 6 omits the performance of the original backbone diffusion models (e.g., SDXL-Turbo, Pixart-DMD). While image quality comparisons with other counting baselines are provided, including the results of the original backbones would strengthen the claim that “D2D yields minimal degradation in image quality” (line 456).

**Questions:**

1. (related to Table 1) Is the proposed method only applicable to single-step diffusion models? If so, is there a specific reason why it cannot be applied to multi-step diffusion models such as SDXL or Pixart-α? Applying the proposed method to SDXL would be particularly valuable, as it would enable direct comparisons with Make It Count.

---

> ### Author Response · Authors · 2025-11-21
> **Rebuttal by Authors**
>
> >W1: The evaluation relies solely on the SOTA counting model CountGD (built upon GroundingDINO), human evaluation should be considered.
>
> **A1:** We thank the reviewer for the insightful comment and have conducted some additional human evaluations as suggested. Our small pilot user study confirms that the proposed method outperforms competing approaches and aligns with the quantitative results obtained using multiple detectors. Below, we describe the evaluation setup in detail. We will update the manuscript with more comprehensive results, contingent on obtaining the appropriate IRB approval, after the rebuttal and before the final version.
>
> Specifically, we sampled 50 prompts from D2D-Small, each of which has five versions of generated images from the SDXL-Turbo and SDXL base models, D2D, Counting Guidance, and TokenOpt, resulting in a total of 250 images. For each study, we included a sample of 30-35 questions, and for each image we asked a short, free-response question of the form "How many [prompted object] are there in the image?" Importantly, we did not disclose which model generated each image. Below is the prompt we used:
>
> *Below are 30 questions. Each question presents an image and a short prompt: "How many <object> are there in the image?" Based on the image, answer the question. Below is an example, to illustrate.*
>
> *All answers must be integers, in numerical format. So 15 is a valid answer, but fifteen is not. 0 is also a possible answer. Emphasis: a single integer response for each question - no additional text. When the answer seems unclear, please make your best effort. Thank you!*
>
> We asked 13 participants to fill out the questionnaire, each consisting of random samples from the pool of paired images and questions.
>
> | Method | Human |
> |-------------|--------------|
> | SDXL-Turbo      |   0.26              |
> | SDXL     |    0.34             |
> | TokenOpt     |    0.26             |
> | Counting Guidance   | 0.18             |
> | D2D     |     **0.48**            |
>
>
>
> >W2: It would be interesting to investigate whether the proposed noise optimization process restricts the diversity of generated images.
>
> **A2:**  We appreciate the valuable input from the reviewer and agree that the tradeoff between overall quality/alignment and diversity has been a long-standing challenge within the T2I community. As suggested, we have conducted a diversity analysis comparing our D2D method with corresponding base models. D2D improves diversity, compared to its corresponding base model. On CoCoCount and D2D-Small, across four seeds. We have added this to Appendix N on lines 1404-1418.
>
> | Diversity Table | Vendi |
> |------|------|
> | SDXL-Turbo | 1.7561059 |
> | SDXL-Turbo + D2D | 1.8231177 |
> | SD-Turbo | 1.8688096 |
> | SD-Turbo + D2D | 1.9801648 |
> | Pixart-DMD | 1.7600942 |
> | Pixart-DMD + D2D | 1.8248777 |
>
>
> >W3: Table 6 omits the performance of the original backbone diffusion models (e.g., SDXL-Turbo, Pixart-DMD), including the results of the original backbones would strengthen the claim that “D2D yields minimal degradation in image quality”.
>
> **A3:** Thank you for the suggestion. For clearer comparison of image quality, we have replaced Table 6 (previously containing four image-quality columns) with a visual plot (Fig. 6a on lines 494-515), including base model performance. We also list the results below for easy reference. In general, D2D improves upon base models in terms of image quality and is comparable to counting baselines.
>
> On CoCoCount:
> | Method | ImageReward |  PickScore |  HPSv2 |  CLIPScore |
> |------------|------------|------------|------------|------------|
> | SDXL-Turbo + D2D (w/ OWLv2) | 1.06 |  23.30 |  0.300 |  32.84 |
> | SDXL-Turbo + D2D (w/ YOLOv9) | 1.06 |  23.33 |  0.300 |  32.88 |
> | SD-Turbo + D2D (w/ OWLv2) | 0.79 |  22.73 |   0.278 |   32.17 |
> | Pixart-DMD + D2D (w/ OWLv2) | 1.17 |  22.73 |  0.292 |   31.85 |
> | SDXL-Turbo | 0.96 |   23.12 |  0.293 |  32.80 |
> | SD-Turbo | 0.72 |  22.59 |  0.277 |  32.54 |
> | Pixart-DMD | 1.18  | 22.82 |  0.291 |  31.79 |
>
> On D2D-Small:
> | Method | ImageReward |  PickScore |  HPSv2 |  CLIPScore |
> |------------|------------|------------|------------|------------|
> | SDXL-Turbo + D2D (w/ OWLv2) |   0.51 | 21.98 |  0.282 |   31.69 |
> | SDXL-Turbo + D2D (w/ YOLOv9) |   0.50 |   22.04 |   0.282   | 31.90 |
> | SD-Turbo + D2D (w/ OWLv2) |   0.21 |   21.46 |  0.259 |  31.04 |
> | Pixart-DMD + D2D (w/ OWLv2) |  0.55  | 21.56 |  0.271 | 31.33 |
> | SDXL-Turbo |   0.40 |  21.98 |   0.279 |  31.89 |
> | SD-Turbo |   0.26 |   21.46 |   0.261   | 31.55 |
> | Pixart-DMD |   0.61 |  21.66   | 0.272  | 31.45 |

---

> ### Author Response · Authors · 2025-11-21
> **Rebuttal by Authors**
>
> >Q1: (related to Table 1) Is the proposed method only applicable to single-step diffusion models? If so, is there a specific reason why it cannot be applied to multi-step diffusion models such as SDXL or Pixart-α? Applying the proposed method to SDXL would be particularly valuable, as it would enable direct comparisons with Make It Count.
>
> **A4:** Thank you for the insightful question. The proposed D2D can be applied to multi-step diffusion models as well; however there is a significant tradeoff in latency. To illustrate this is possible, we applied the proposed D2D critic to Direct Noise Optimization [a], an initial noise optimization framework for multi-step models. We found that convergence takes approximately 7 minutes. The results have been added to Appendix M on lines 1374-1395.
>
> -------
> [a] Inference-Time Alignment of Diffusion Models with Direct Noise Optimization. ICML '25.

---

### Official Review · Reviewer_NJkJ · 2025-11-04

**Soundness:** 2
**Presentation:** 3
**Contribution:** 3
**Rating:** 6
**Confidence:** 4

**Summary:**

This paper addresses numeracy in text-to-image models through reward-based inference-time optimization, making two main contributions. First, they design a differentiable counting reward $\mathcal{L}_{D2D}$ derived from object detectors, which are more accurate than regression-based counters in low-density scenarios but previously unusable as rewards due to non-differentiable enumeration. They achieve this through custom activation functions that convert detector logits into gradient-friendly signals. Second, they propose the Latent Modifier Network (LMN), a lightweight 3-layer MLP that transforms the initial noise for reward optimization. Unlike prior work that directly tunes the noise, the LMN provides a larger parameter space while preserving portions of the original noise through a weighted mixing scheme. Experiments show consistent improvements in counting accuracy.

**Strengths:**

- **Novel reward function design:** Converting non-differentiable detectors into differentiable counting rewards through steep sigmoids and logit scaling (Eq. 1-2) addresses a limitation where detectors outperform regression models in low-density counting but couldn't previously be used for gradient-based optimization.
- **Two complementary contributions:** The differentiable detector-based reward ($\mathcal{L}_{D2D}$) and the LMN architecture for reward optimization. Table 5 shows the LMN improves numeracy by 10% points over direct noise optimization when using the same reward, suggesting the LMN is a valuable contribution independent of the specific reward function.
- **Consistent empirical results:** Consistent improvements across multiple base models and settings in reasonable time-frames.

**Weaknesses:**

- **Limited Discussion of General-Purpose Application**: The methodology is tailored for prompts containing numerical targets, but the paper does not discuss how the framework should behave with general, non-numeric prompts. It is unclear whether the D2D optimization is intended to be selectively activated, or what the default behavior might be in the absence of a numerical target (e.g., "few", "some", or "many"). How about very long complex prompts with lots of content. Additionally, it's not really clear what the effect of D2D is on aesthetics and diversity of generated images.
- **Justification for Reward Function Design Choices**: While the activation function design is intuitive, several design choices lack theoretical justification or empirical comparison to alternatives:
	- **Scaling formulation (Eq. 2):** The justification for multiplying by $(z_i - \tau_z)$ to combat sigmoid plateauing is limited. Did the authors explore other options, such as exponential scaling, learned scaling functions, or other monotonic transformations that could provide stronger gradients? The paper does not compare the chosen formulation against alternatives.
	- **Mixing weight:** Table 9 shows $w=0$ achieves the best numeracy but produces "patchy visual artifacts," suggesting a tradeoff between reward optimization and image quality. The basis for choosing $w=0.2$ could be better explained. More detailed analysis of this numeracy vs. image quality tradeoff in the main paper would be valuable.
	- **Noise regularization sharpening**: The paper proposes a much more sharped version of previous regularization in noise space by exponentiating it by a power of 10. To me it's unclear why this change was made, and not really motivated in the paper.
- **LMN Contribution Not Fully Explored**: The LMN appears to be a contribution independent of the counting reward (Table 5 demonstrates improvements over direct noise tuning), yet it receives limited investigation.
	- It is unclear whether the LMN improves optimization of other rewards beyond counting (e.g., ImageReward or similar like ReNO). Currently, Table 5 only compares LMN vs. direct noise tuning using $\mathcal{L}_{D2D}$. Evaluation with other rewards would help establish whether the LMN is a general contribution to inference-time reward optimization or specific to counting tasks.
	- The choice of a 3-layer MLP is not justified through ablations. It would be informative to explore variations such as 2 or 4 layers, different widths, or alternative architectures.

Minor Issues:
- Including the base model performance in Table 6 would facilitate easier comparison of how the reward-based optimization affects image quality metrics.
- D2D is proposed as a differentiable surrogate for the non-differentiable detector count, but no analysis validates this approximation. It would be informative to analyze how well $\mathcal{L}_{D2D}$ correlates with the actual detector count $D(I)$ throughout the optimization trajectory, and under what conditions the surrogate diverges from the true count (e.g., when many bboxes cluster near threshold $\tau_z$).

**Questions:**

- How does the LMN perform for human-preference rewards, e.g. ReNO?
- What's the effect of D2D on the diversity of generated images?
- What is the motivation behind the sharper noise regularization? How does it imact diversity?
- What's the effect of D2D on general image quality? This could be made more clear through adding baseline performances to Table 6
- Could D2D be also used to fine-tune a T2I model, e.g. with Adjoint Matching?
- Are there scenarios where the surrogate diverges from the true count?
- How many optimization steps are used? How important is the pre-inference alignment stage?

---

> ### Author Response · Authors · 2025-11-21
> **Rebuttal by Authors**
>
> >W1: Limited discussion of general-purpose application for general non-numeric and long complex prompts.
>
> **A1:** We appreciate the reviewer’s thoughtful feedback. Our motivation for focusing on numerical prompts with clear object-count pairs is that they provide a well-defined optimization scenario and measurable evaluation criteria.
>
> We understand, however, that many prompts in practice do not specify explicit numbers or could be complex to parse for T2I models. To address these cases, we performed a pilot study using in-context prompting with an LLM to distinguish numerical prompts and extract relevant object-count information from long and complex prompts when available. For non-numeric terms like “few” or “many,” which lack strict grounding, we believe it is reasonable to introduce flexibility by heuristically mapping these words to approximate numbers where appropriate. Likewise, the framework can leverage LLMs to filter out objects or prompts that are inherently uncountable or unsuitable for numeric interpretation (e.g., “water” in “some water”).
>
> In our updated manuscript, Appendix J (lines 1188–1278) details our prompt instructions and example outcomes:
>
>  - Non-numerical case: ChatGPT identifies “A photo of some cups” as a numerical prompt and reasonably chooses “3” as a target.
>
> - Complex prompt case: It recognizes “A magnificent landscape with lots of sunshine, animals everywhere, and plenty of greenery.” as non-numeric.
>
> In summary, our D2D framework is designed to be adaptable: it can be activated for numerical prompts and rely on LLM-based filtering and heuristics to handle more general or complex content.

---

> ### Author Response · Authors · 2025-11-21
> **Rebuttal by Authors**
>
> >W2: Justification for reward function design choices on the scaling formulation, mixing weight, and sharper noise regularization.
>
> **A2:** We thank the reviewer for the valuable questions. Below, we present additional experiments and analysis as an elaborated justification for our modeling designs.
>
> 1. **Scaling formulation.** We conducted additional experiments with other alternative exponential scaling formulations listed below on CoCoCount, seed=0 on SDXL-Turbo, and demonstrate that our current formulation achieves better numeracy and faster inference time. We believe this is attributed to the shape of the curvature in domains of interest. The idea is that we want to design a fully-smooth loss that will effectively update each bbox's logit in order to bring the total count closer to the requested count. To illustrate, if the total generated count is bigger than requested count, then the overall objective is to bring the logits down. However, practically speaking, on the level of individual logits, this means focusing on bringing down the logits that are above the $\tau$ and allowing some flexibility for logits that are already below $\tau$ so that the optimization can focus on the logits greater than $\tau$. This points to the motivation that we want a strong gradient signal above $\tau$ and little signal below. With this in mind, we added to the appendix Figure 14 on lines 1316-1346 plotting the curves of the first three scaling formulations from the table, illustrating how they behave above and below $\tau$. Our formulation has a strong signal above and weak signal below $\tau$. $e^{z-\tau}$ still maintains a significant gradient below $\tau$. $e^{\frac{1}{1+e^{\beta(z-\tau)}}}$ still shows plateaus above $\tau$. We have updated Appendix K (starting on line 1296) with these details.
>
> | Scaling formulation                                      | Numeracy | Inference time (s) |
> |--------------------------------------------------------------|--------------|---------------------------|
> | $\frac{1}{1+e^{-\beta(z-\tau)}} \cdot (z-\tau)$ (ours)  |   **55.5**       |  **17.10**                 |
> | $e^{z-\tau}$                                                      |    52.0      | 56.28                 |
> | $e ^ \sigma(\beta(z - \tau))$                             |    31.5       |    32.24              |
> | $z-\tau$                             |    50.5       |    53.23             |
> | $ln(1+e^{z-\tau})$                             |    55.0       |    20.09             |
> | $(z-\tau) \cdot e^{z-\tau}$                             |    49.0       |    72.72             |
> | $(z-\tau) \cdot ln(1+e^{z-\tau})$                             |    54.0       |    22.31             |
> | $\sigma(\beta(z-\tau)) \cdot e^{z-\tau}$                             |    50.5      |    63.96             |
> | $\sigma(\beta(z-\tau)) \cdot ln(1+e^{z-\tau})$                             |    55.0       |    18.65    |
>
> 2. **Mixing weight.** We select $w=0.2$ because it provides the best balance between numeracy and visual quality. As we have additionally clarified in Appendix I.1 (lines 1029-1062), setting $w=0$ achieves the strongest numeracy but introduces patch-like artifacts (Fig. 11, column 1). Using $w=0.2$ substantially reduces these artifacts while retaining high numeracy performance (Fig. 11, column 2). Larger values of $w$ increase the contribution of the original Gaussian noise, which acts as an additional form of regularization but weakens numeracy optimization.
>
> 3. **Sharper noise regularization.** Intuitively, our sharper regularization term aims to sharpen the gradient for larger deviations and flattens the gradient for smaller deviations from the minimum. We have added Figure 13 in Appendix I.3 (lines 1086-1103), plotting the curves of $x^2$ and $x^{10}$ to illustrate. Relative to $x^2$, $x^{10}$ appears more flat around the minimum and steeper away from the minimum, effectively allowing the optimization process to focus more on *numeracy* optimization when the latent is already well-regularized. Our ablation analysis confirms that this variant of the regularization term leads to overall numeracy improvements across both CoCoCount and D2D-Small, as seen in the table below (avg. over four seeds on SDXL-Turbo). We have added these details, figures, and table to Appendix I.3 (starting at line 1080).
>
> | Method              | CoCount | D2D-Small |
> |--------------------------------|-------------------|-------------------|
> | D2D w/ original regularization   |        53.88        |     42.44         |
> | D2D w/ sharper regularization    |        **55.62**        |     **43.69**       |

---

> ### Author Response · Authors · 2025-11-21
> **Rebuttal by Authors**
>
> >W3: LMN contribution not fully explored: It is unclear whether the LMN improves optimization of other rewards beyond counting; the choice of a 3-layer MLP is not justified through ablations.
>
> **A3:** We appreciate the insightful suggestions.
>
> 1. We conducted experiments using LMN on the image quality metrics adopted by ReNO. Based on our preliminary results, LMN does not yield significant improvement on these metrics compared to its positive impact on numeracy.
>
> That said, the LMN does encourage a more stable optimization process due to its mixing weight—which preserves a consistent Gaussian component in the noise originating from the original noise, which acts as another form of regularization—and adaptive scaling technique. Specifically, we conducted experiments on 200 prompts from CoCoCount and found that ReNO encountered explosive gradients and remained stuck on 95 out of 200 prompts, whereas with D2D, while we encountered non-finite gradient norms on those prompts, we were able to recover in every instance via adaptive scaling.
>
> 2. For the choice of 3-layer MLP, we performed additional ablation studies and present the results below. Between MLP architectures, the 3-layer, 100-width and 200-width variants perform the best, but the former is more parameter-efficient. We also tried a 6-layer CNN (with adjusted kernels and channels to approximate the parameter count of the 3-layer, 100-width MLP), which yielded a slight 1% boost in numeracy. Our results demonstrate that even a small and simple MLP works quite well as an LMN architecture. Evaluated on 200 prompts from CoCoCount on SDXL-Turbo.
>
> | Variant |  Numeracy |
> |-----------|------------------|
> 3 layers, width 100 | 55.5 |
> 2-layers, width 100 |  52.5|
> 4-layers, width 100 | 52.0  |
> 3-layers, width 50 |  54.5 |
> 3-layer, width 200 | 55.5  |
> CNN (w/ approximately same number of parameters)  | 56.5  |
>
> We have integrated the above experimental results and analysis in our updated manuscript, in Appendix I.4, on lines 1134-1168.
>
> >W4: Including the base model performance in Table 6 would facilitate easier comparison of how the reward-based optimization affects image quality metrics.
>
> **A4:** Thank you for the suggestion. For clearer comparison of image quality, we have replaced Table 6 (previously containing four image-quality columns) with a visual plot (Fig. 6a on lines 494-515), including base model performance. We also list the results below for easy reference. In general, D2D improves upon base models in terms of image quality and is comparable to counting baselines.
>
> On CoCoCount:
> | Method | ImageReward |  PickScore |  HPSv2 |  CLIPScore |
> |------------|------------|------------|------------|------------|
> | SDXL-Turbo + D2D (w/ OWLv2) | 1.06 |  23.30 |  0.300 |  32.84 |
> | SDXL-Turbo + D2D (w/ YOLOv9) | 1.06 |  23.33 |  0.300 |  32.88 |
> | SD-Turbo + D2D (w/ OWLv2) | 0.79 |  22.73 |   0.278 |   32.17 |
> | Pixart-DMD + D2D (w/ OWLv2) | 1.17 |  22.73 |  0.292 |   31.85 |
> | SDXL-Turbo | 0.96 |   23.12 |  0.293 |  32.80 |
> | SD-Turbo | 0.72 |  22.59 |  0.277 |  32.54 |
> | Pixart-DMD | 1.18  | 22.82 |  0.291 |  31.79 |
>
> On D2D-Small:
> | Method | ImageReward |  PickScore |  HPSv2 |  CLIPScore |
> |------------|------------|------------|------------|------------|
> | SDXL-Turbo + D2D (w/ OWLv2) |   0.51 | 21.98 |  0.282 |   31.69 |
> | SDXL-Turbo + D2D (w/ YOLOv9) |   0.50 |   22.04 |   0.282   | 31.90 |
> | SD-Turbo + D2D (w/ OWLv2) |   0.21 |   21.46 |  0.259 |  31.04 |
> | Pixart-DMD + D2D (w/ OWLv2) |  0.55  | 21.56 |  0.271 | 31.33 |
> | SDXL-Turbo |   0.40 |  21.98 |   0.279 |  31.89 |
> | SD-Turbo |   0.26 |   21.46 |   0.261   | 31.55 |
> | Pixart-DMD |   0.61 |  21.66   | 0.272  | 31.45 |
>
>
> >W5: It would be informative to analyze how well $L_{D2D}$ correlates with the actual detector count and under what conditions the surrogate diverges from the true count.
>
> **A5:** Thank you for the question. We would like to clarify that Equation 1 (line 207), which is the sum of sigmoids, computes the soft detector count. On the other hand, $L_{\text{D2D}}$ (Equation 2 on lines 209-210) is the convergence-friendly adaptation of Equation 1, where each sigmoid is scaled by the corresponding logits before summation. As a result of this per-bounding-box logit scaling by $z-\tau$, the $L_{\text{D2D}}$ is heavily influenced by the magnitude of individual logits themselves. For example, despite the same detector count, if the bounding box confidence scores are higher, then the corresponding term in the D2D loss will be larger. So $L_{\text{D2D}}$ is not directly correlated with the raw detector count.
>
> We have added Figure 15 (lines 1353-1371) to Appendix L, which plots a sample trajectory of $L_{\text{D2D}}$ and the soft detector count over numeracy optimization steps. While their absolute scales and stepwise behavior differ, their coarse trends correlate.

---

> ### Author Response · Authors · 2025-11-21
> **Rebuttal by Authors**
>
> >Q1: How does the LMN perform for human-preference rewards, e.g. ReNO?
>
> **A6:** Thank you for the question. We address this in our above response A3.
>
> >Q2: What's the effect of D2D on the diversity of generated images?
>
> **A7:** Thank you for the question. We conducted additional diversity evaluations, using the Vendi metric [c]. D2D improves diversity, compared to its corresponding base model, whilst being comparable to counting/enhancement baselines. The table below shows Vendi scores computed on CoCoCount and D2D-Small across four seeds. We have added this to the paper in Appendix N on lines 1407-1413.
>
> | Method | Vendi |
> |------|------|
> | SDXL-Turbo | 1.7561059 |
> | SDXL-Turbo + D2D | **1.8231177** |
>
> >Q3: What is the motivation behind the sharper noise regularization? How does it impact diversity?
>
> **A8:** Thank you for the question. For motivation behind sharper regularization, please refer to A2. We conducted additional diversity evaluations, using the Vendi metric, on CoCoCount and D2D-Small on the base model SDXL-Turbo across four seeds. The sharper variant improves diversity, compared to ReNO's variant, as shown in the table below. We have added this to the paper in Appendix I.3 on lines 1127-1133.
>
> | Method | Vendi |
> |------|------|
> | D2D w/ original reg | 1.8198230 |
> | D2D w/ sharper reg  | **1.8231177** |
>
> >Q4: What's the effect of D2D on general image quality? This could be made more clear through adding baseline performances to Table 6.
>
> **A9:** Thank you for the question. We have reformatted Table 6 as a figure for better visualization and comparison, and added the base model performance as suggested. Please refer to A4.
>
> >Q5: Could D2D be also used to fine-tune a T2I model, e.g. with Adjoint Matching?
>
> **A10:** Thank you for the insightful question. In our proposed numeracy generation scenario, both inference-time optimization and fine-tuning approaches aim to steer the base distribution towards a target tilted distribution defined by a reward model that captures human preferences; however, the specific implementation can vary across different component designs in our D2D framework.
> 1. At a fundamental level, the loss function in D2D does not fundamentally differ from other gradient-based objectives used for fine-tuning. Thus, the differential loss in our method can also be applied to fine-tuning.
> 2. On the implementation side, we observe practical advantages of D2D in inference-time optimization over fine-tuning. In particular, fine-tuning the entire base distribution to achieve general precise counting tends to be more challenging than steering the output towards a specific prompt with a precise count and object labels.
> 3. Finally, regarding the referenced work Adjoint Matching [a], while the D2D loss is not strictly orthogonal to its core idea, which factorizes the boundary distributions as independent, the Latent Modifier Network (LMN) in D2D takes a complementary approach by leveraging the implicit noise prior for improved numeracy alignment. From this perspective, LMN is more closely related to another line of work that leverages this implicit dependency between the Gaussian prior and target distribution for better control and alignment, such as [b].
>
> From a more general perspective, inference-optimization offers flexibility of preserving the generalization ability of base models without altering the pre-trained weights.
>
> >Q6: Are there scenarios where the surrogate diverges from the true count?
>
> **A11:** Thank you for the questions. Please refer to A5.

---

> ### Author Response · Authors · 2025-11-21
> **Rebuttal by Authors**
>
> >Q7: How many optimization steps are used? How important is the pre-inference alignment stage?
>
> **A12:** Thank you for the question. We used a capped number of optimization steps at 200 for D2D-Small and CoCoCount and at 400 for the more challenging D2D-Large/Multi benchmarks. For SDXL-Turbo and SD-Turbo on D2D-Large, we actually found that setting the maximum to 200 was sufficient, so we used a lower cap for these. Because we perform early stopping, the optimization process usually converges well under 200 steps. For example, on CoCoCount, early stopping occurred within 120 steps for 127 out of 200 prompts (seed=0, on base model SDXL-Turbo).
>
> We conduct additional experiments without the pre-inference alignment stage and report below. On CoCoCount, SDXL-Turbo, seed=0, we find that without pre-inference alignment, it generally takes an additional 20-30 steps during inference time to align the network to Gaussian outputs, which effectively reduce the number of steps available to perform numeracy optimization, because we cap the max number of steps. We have added this to Appendix I.5 on lines 1162-1168.
>
> | | Numeracy |
> |------|------|
> | w/ pre-inference alignment | **55.5** |
> | w/o pre-inference alignment | 47.0  |
>
>
> ----
> [a] Adjoint Matching: Fine-tuning Flow and Diffusion Generative Models with Memoryless Stochastic Optimal Control, ICLR’25
>
> [b] The Silent Assistant: NoiseQuery as Implicit Guidance for Goal-Driven Image Generation, ICCV’25
>
> [c] The Vendi Score: A Diversity Evaluation Metric for Machine Learning. TMLR. 2023.

---

### Author Response · Authors · 2025-11-21
**Overall Responses and Summary of Revisions**

First, we would like to thank all reviewers for their valuable input and for their recognition of our work on the novel design of the differential counting critic (R-NJkJ), clear motivation and presentation (R-Jery, R-LUcL), in-depth analysis (R-Jery), and consistent performance improvement (R-NJkJ, R-Jery, R-LUcL, and R-Zbh8). We also greatly appreciate the valuable suggestions, such as further in-depth quality and diversity analysis (R-NJkJ, R-Jery); additional comprehensive evaluation via human studies (R-Jery), as well as a generalization study on both multi-step (R-Jery) and larger one-step models (R-LUcL).

We have conducted extensive additional experiments during the rebuttal period, including:
- Diversity analysis using Vendi scores
- Pilot human studies
- Ablation studies on LMN architecture variants
- Experiments with alternative scaling formulations
- Analysis of D2D loss correlation with soft detector counts
- Numeracy evaluations on newer/larger models

These results are highlighted in the updated manuscript and supplementary material (appendix). We hope this could be a more comprehensive version after integrating all valuable inputs.

---

### Meta-Review · Area_Chair_2zSq · 2026-01-07

**Summary:**

The concerns can be summarized as below:
1. Limited Discussion of General-Purpose Application on how to handle 'many', 'few' in the prompt and how to handle the long complex prompt
2. Justification for Reward Function Design Choices
3. The LMN seems not very related to the other part of the paper.
4. Whether the proposed approach restrict the diversity of the generated images.
5. Didn't compare with more recent work on the T2I domain.

**Reviewer Concerns:**

1. For the more general-purpose application, the author's rebuttal mentioned the LLM can be leveraged to extend the prompt that removing the 'many' or 'few' with a more concrete number. This is actually aligned with the prompt expansion idea. Frankly speaking, this is a very smart way to address this concern. But this relies on the LLM to generate a meaningful number. If the LLM generated a super weird number (some number doesn't make sense), we are not sure how the proposed approach would behave on that.
2. The author rebuttal provided experiments to justify the reward function design choice.
3. The author rebuttal demonstrated that the LMN does not yield significant improvement on these metrics compared to its positive impact on numeracy. I think the rebuttal address the concern for the relevance of the LMN and the rest of paper.
4. The rebuttal demonstrates that the D2D can improve the diversity.
5. The author's rebuttal demonstrated that with the more recent work, the proposed SDXL+D2D still outperforms them on Numeracy.

**Reviewer Scores:**

This is a borderline paper. I thank the reviewer comments and the author for the comprehensive rebuttal.
After reading the review and the rebuttal, I still have concerns with the proposed approach.
1. On the proposed approach improving the numeracy part, given the SDXL is a quite out-dated baseline, I wonder whether the proposed approach truly improve the counting. For example, the figure 5 first row right most image (the one that D2D corrected the over-generation), the dogs are not generated correctly (e.g. the limbs are not in the right place, the faces are incorrect.) For this generated results, I wonder whether this is considered to be generated correctly in terms of counting? More fundamentally, the model might generate the correct number of objects, but all those objects are not rendered correctly. I wonder whether we need to make that concession? But the challenging side of addressing this concern is the SoTA image generation model are mostly close-sourced. It is hard to conduct a thorough study on that.
2. I am slightly surprised that the D2D can improve the diversity as shown in the rebuttal. The author might need to provide more conceptual reasoning for studying this effect.

Given that, I like the proposed approach. Therefore, I provided the confidences as This decision can be bumped up.

---

### Decision · Program_Chairs · 2026-01-26

Reject